# The acinar differentiation determinant PTF1A inhibits initiation of pancreatic ductal adenocarcinoma

Nathan M Krah[1], Jean-Paul De La O[1], Galvin H Swift[2], Chinh Q Hoang[2], Spencer G Willet[3], Fong Chen Pan[3], Gabriela M Cash[1], Mary P Bronner[4], Christopher VE Wright[3], Raymond J MacDonald[2], L Charles Murtaugh[1]*

[1]Department of Human Genetics, University of Utah, Salt Lake City, United States; [2]Department of Molecular Biology, University of Texas Southwestern Medical Center, Dallas, United States; [3]Department of Cell and Developmental Biology, Vanderbilt University Medical Center, Nashville, United States; [4]Department of Pathology, Huntsman Cancer Hospital, University of Utah, Salt Lake City, United States

**Abstract** Understanding the initiation and progression of pancreatic ductal adenocarcinoma (PDAC) may provide therapeutic strategies for this deadly disease. Recently, we and others made the surprising finding that PDAC and its preinvasive precursors, pancreatic intraepithelial neoplasia (PanIN), arise via reprogramming of mature acinar cells. We therefore hypothesized that the master regulator of acinar differentiation, PTF1A, could play a central role in suppressing PDAC initiation. In this study, we demonstrate that PTF1A expression is lost in both mouse and human PanINs, and that this downregulation is functionally imperative in mice for acinar reprogramming by oncogenic KRAS. Loss of *Ptf1a* alone is sufficient to induce acinar-to-ductal metaplasia, potentiate inflammation, and induce a KRAS-permissive, PDAC-like gene expression profile. As a result, *Ptf1a*-deficient acinar cells are dramatically sensitized to KRAS transformation, and reduced *Ptf1a* greatly accelerates development of invasive PDAC. Together, these data indicate that cell differentiation regulators constitute a new tumor suppressive mechanism in the pancreas.

*For correspondence: murtaugh@genetics.utah.edu

**Competing interests:** The authors declare that no competing interests exist.

## Introduction

Pancreatic ductal adenocarcinoma (PDAC) has a dismal prognosis, with a 5-year survival rate around 5% and a cure rate approaching zero. The most up-to-date chemotherapy regimens extend life only minimally (*Ryan et al., 2014*), and patients undergoing resection of ostensibly local tumors almost invariably succumb to recurrent disease. While this observation suggests that PDAC is usually metastatic at the time of diagnosis, recent studies suggest that tumors require over 20 years to evolve from precancerous pancreatic intraepithelial neoplasia (PanIN) to invasive carcinoma (*Yachida et al., 2010*). Thus, in principle, there is a large window of time for effective and early detection, prevention, and treatment, provided appropriate methods are in place. Therefore, defining the cell type of origin and characterizing the process of PanIN-PDAC evolution within the physiologic context of key risk factors (e.g., chronic pancreatitis, type 2 diabetes, or genetic cancer syndromes [*Ryan et al., 2014*]) is crucial to finding effective therapies.

The vast majority of human PanINs and PDAC contain activating mutations in the *KRAS* oncogene, which have been shown in mice to represent driver mutations for PanIN initiation, maintenance, and progression to PDAC (reviewed in *Pasca di Magliano and Logsdon, 2013*). Notably, the progression of PanINs to PDAC is accompanied by additional mutations in tumor suppressor genes, such as *INK4A, CDKN2A, TRP53* (commonly referred to as *P53*), and *DPC4/SMAD4* (*Ryan et al., 2014*). While

**eLife digest** Pancreatic cancer is one of the most lethal forms of cancer, with fewer than 20% of people surviving for longer than twelve months after diagnosis. Two types of genetic mutation play important roles in pancreatic cancer. First, genes called oncogenes can be activated by mutations to drive unscheduled cell division. Second, the genes for tumor suppressors—proteins that prevent cells from dividing when they should not—can be switched off due to other mutations. Together, these mutations cause cells to over-proliferate and disrupt the structure of the pancreas.

In a healthy pancreas, several different cell types perform various roles: acinar cells produce proteins that digest food, ductal cells carry these proteins to the intestine, and β cells produce insulin. Certain proteins are responsible for telling each of these cells what tasks to perform, which defines their so-called differentiation state. The protein PTF1A is crucial for establishing the differentiation state of acinar cells. In the most common form of pancreatic cancer, acinar cells are reprogrammed to become ductal cells. Moreover, pancreatic cancer cells contain much lower levels of PTF1A than normal pancreatic cells.

To explore the connection between PTF1A and pancreatic cancer, Krah et al. deleted the gene for PTF1A in mice. This led to acinar cells being reprogrammed to become ductal cells. Additionally, when an oncogene mutation was activated at the same time as the gene for PTF1A was deleted, Krah et al. observed the rapid formation of large numbers of malignant pancreatic tumors in the mice. PTF1A therefore protects against pancreatic cancer by acting as a tumor suppressor and keeping acinar cells in their healthy, differentiated state.

Unlike other tumor suppressors, however, PTF1A levels are reduced in cancer cells by a mechanism that does not involve a genetic mutation. Therefore, a future challenge is to determine how the amount of PTF1A protein is reduced, and in the longer term, to explore if it is possible to reverse cancer progression by forcing cancer cells back into their original differentiation state.

the initiation and progression of PDAC has understandably been difficult to study in human patients or to model in human tissue (*Boj et al., 2015*), much has been learned from the 'KC' mouse model in which a <u>C</u>re-inducible oncogenic <u>K</u>ras allele ($Kras^{LSL-G12D}$) causes focal PanIN formation when activated universally in the pancreas (*Aguirre et al., 2003*; *Hingorani et al., 2003*; *Murtaugh, 2014*). Until recently, PDAC was primarily thought to originate from pancreatic ductal cells because of the cancer's duct-like epithelial phenotype. However, recent studies indicate that PanINs (*De La et al., 2008*; *Habbe et al., 2008*; *Kopp et al., 2012*) and PDAC (*Ji et al., 2009*) can be initiated by activating oncogenic $Kras^{G12D}$ expression specifically within mature acinar cells, while $Kras^{G12D}$ activation in adult duct cells or centroacinar cells has little or no effect. Interestingly, even in the KC mouse model, where embryonic Cre recombinase activity directs $Kras^{G12D}$ expression to nearly every cell of the mature pancreas, only a small number of acinar cells eventually give rise to PanINs. The mechanism by which most acinar cells remain refractory to $Kras^{G12D}$-mediated transformation has not been elucidated. An attractive hypothesis is that the factors that induce and maintain acinar cell differentiation state play a crucial role in inhibiting the acinar cell reprogramming step that serves to initiate PDAC formation and progression (*Rooman and Real, 2012*; *Bailey et al., 2014*; *Murtaugh, 2014*).

Consistent with acinar cells as the cell of origin in PDAC, and acinar cell identity being a protective mechanism against $Kras^{G12D}$-mediated transformation, recent genome-wide association studies identified PDAC risk-associated single-nucleotide polymorphisms in the non-coding region of the gene encoding the acinar differentiation transcription factor *NR5A2*, also known as *LRH-1* (*Petersen et al., 2010*). These findings have been confirmed in mouse studies, where pan-pancreatic loss of *Nr5a2* significantly sensitizes pancreatic cells to *KRAS*-induced PanIN initiation. Additionally, pancreatic *Nr5a2* is necessary to regenerate the acinar compartment following caerulein-induced pancreatitis (*Flandez et al., 2014*; *von Figura et al., 2014b*). These studies begin to define how acinar cell differentiation programs may act as an important defense in a progressively severe sequence of events: loss of the mature acinar phenotype, PanIN initiation, and formation of PDAC.

In adult pancreata, NR5A2 maintains acinar cell identity by cooperating with the acinar-specific pancreas-specific transcription factor 1 (PTF1) complex, which has binding motifs upstream of essentially all acinar differentiation products, such as *Cpa1*, *Cela1*, and *Cel* (*Holmstrom et al., 2011*).

The central specificity component of PTF1 is the cell type-restricted basic helix-loop-helix protein, PTF1A (also known as p48). PTF1A plays two distinct roles during pancreatic organogenesis. First, it is necessary for the growth and morphogenesis of the early pancreatic epithelium, working to impart multipotency and second, its upregulation and lineage-specific interaction with RBPJL promotes acinar differentiation and regulates acinar cell-specific gene expression in adulthood (*Krapp et al., 1998*; *Rose et al., 2001*; *Kawaguchi et al., 2002*; *Masui et al., 2007, 2010*; *Holmstrom et al., 2011*). Homozygous mutations in human *PTF1A* that disrupt its function or expression cause pancreatic agenesis, supporting its role in pancreas development (*Sellick et al., 2004*; *Weedon et al., 2014*). The severity of this phenotype, however, precludes analysis of PTF1A function in mature human acinar cells. Importantly, in the adult pancreas, PTF1A drives its own expression and that of other PTF1 components via a positive autoregulatory loop (*Masui et al., 2008*). Consistent with the central role of this transcription factor in defining and maintaining acinar cell identity, we have shown that PTF1A is downregulated in acinar cells transformed by *Kras*$^{G12D}$ and Notch activation (*De La et al., 2008*). Beyond these observations, however, a definitive role of PTF1A in regulating the pathogenesis of PDAC and other adult pancreatic pathology has not yet been described. Based on the studies described above, we hypothesized that loss of PTF1A is a necessary and sufficient step in acinar cell reprogramming, the initiation of PanINs, and the progression of PDAC.

In this study, we demonstrate that downregulation of PTF1A is a decisive and rate-limiting step in acinar-to-ductal metaplasia (ADM), PanIN initiation, and PDAC progression. Our findings suggest that PTF1A acts in a dosage-sensitive manner to safeguard the pancreatic acinar population against both oncogene activity and environmental insults, such as damage caused by pancreatitis. Our study is the first to establish that an endogenous, autoregulatory differentiation program protects mature pancreatic cells from cancer initiation.

## Results

### PTF1A expression is lost during KRAS-induced transformation of acinar cells and in human PanINs

We have previously demonstrated that *Ptf1a* expression is lost when activated Notch and *Kras*$^{G12D}$ work synergistically to reprogram acinar cells into PanINs (*De La et al., 2008*). Given that *Ptf1a* is a central regulator of acinar cell gene expression, we hypothesized that this transcription factor should also be downregulated when acinar cells are transformed by oncogenic *Kras*$^{G12D}$ alone, as well as in human PanINs. To test this hypothesis, we activated *Kras*$^{G12D}$ specifically in acinar cells using a tamoxifen-inducible Cre expressed by the endogenous *Ptf1a* locus (*Ptf1a*$^{CreERT}$) (*Kopinke et al., 2012*; *Pan et al., 2013*). Like the widely used *Ptf1a*$^{Cre}$ allele (*Kawaguchi et al., 2002*), *Ptf1a*$^{CreERT}$ is a 'knock-in/knock-out' allele, and therefore, these mice are functionally heterozygous for *Ptf1a*. We induced *Kras*$^{G12D}$ expression at 6 weeks of age and harvested pancreata 9 months later. While most acini appeared histologically normal and resistant to KRAS-mediated transformation (*Figure 1A*), there was intermittent PanIN formation throughout the pancreas (*Figure 1B*), as previously reported (*Kopp et al., 2012*). By immunohistochemistry (IHC), normal acinar cells in these tissues exhibited robust nuclear PTF1A (*Figure 1C*); however, PTF1A was strongly decreased or absent in all acinar-derived PanIN lesions (*Figure 1D*). To extend these studies to human pancreatic cancer initiation, we stained pathological specimens (n = 4) containing both normal acinar tissue (*Figure 1E*) and PanIN lesions (*Figure 1F*). As observed in the *Kras*$^{G12D}$ mouse model, normal acini exhibited a strong PTF1A nuclear signal (*Figure 1G*), but PTF1A was largely absent from epithelial cell nuclei within PanINs (*Figure 1H*). In a small fraction of human PanINs, low levels of PTF1A were observed in a subset of epithelial cells (*Figure 1—figure supplement 1*). Residual PTF1A expression is consistent with the finding that approximately one-third of human PDAC samples express low levels of acinar-specific genes (*Collisson et al., 2011*).

### Deletion of *Ptf1a* causes acinar-ductal metaplasia and dramatically enhances KRAS-driven acinar cell transformation

In order to determine whether PTF1A downregulation was a functionally important step in PanIN initiation, or a side effect of acinar cell transformation itself, we used an inducible system to delete *Ptf1a* both in the absence and presence of oncogenic *Kras*$^{G12D}$. In this model, we combined the *Ptf1a*$^{CreERT}$ allele, which does not express PTF1A protein, with a 'floxed' *Ptf1a* allele, to generate *Ptf1a*

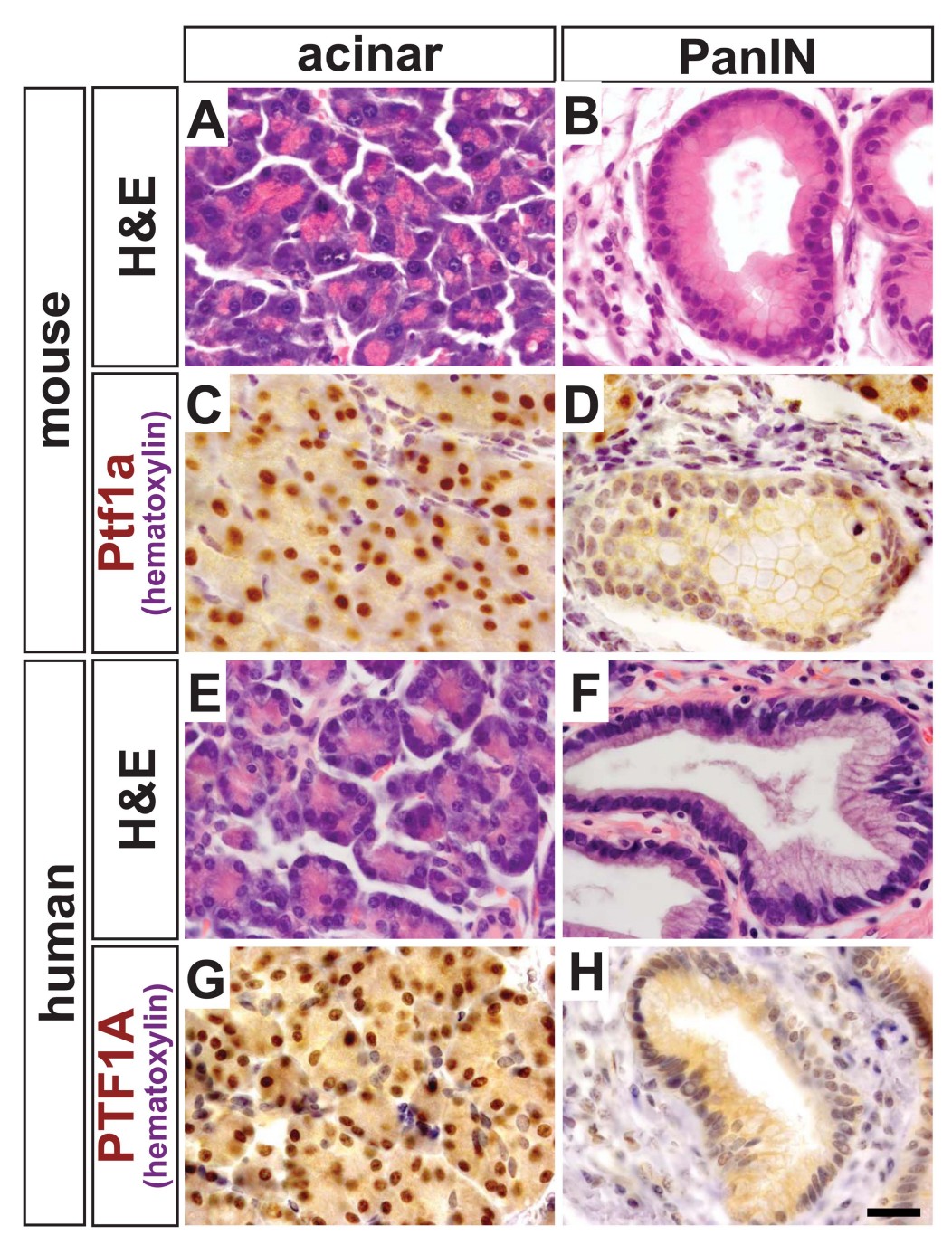

**Figure 1**. PTF1A is downregulated in PanINs from mice and humans. (**A**, **B**) H&E staining of normal acinar and pancreatic intraepithelial neoplasia (PanIN) tissue of $Ptf1a^{CreERT}$; $Kras^{LSL-G12D}$ pancreata. (**C**, **D**) PTF1A immunohistochemistry (IHC) of mouse acinar and PanIN tissue. (**E**, **F**) H&E staining of human acinar and PanIN tissue. (**G**, **H**) PTF1A immunostaining of normal acinar and PanIN tissue of human. Scale bar: 25 μm.

The following figure supplement is available for figure 1:

**Figure supplement 1**. PTF1A expression in rare epithelial cells of human PanINs.

conditional knock-out (cKO) mice of the genotype *Ptf1a*[CreERT/lox]. We also crossed *Kras*[LSL-G12D] onto this *Ptf1a* cKO background. Negative control littermates were *Ptf1a* heterozygous (*Ptf1a*[CreERT/+]) without oncogenic *Kras*. An additional control group, representing baseline PanIN initiation in the presence of wild-type PTF1A, consisted of *Ptf1a*[CreERT/+]; *Kras*[LSL-G12D] littermates (henceforth referred to as *Kras*[G12D] mice). All inducible-Cre mice also contained a *R26R*[EYFP] reporter (*Srinivas et al., 2001*), which allowed monitoring of the frequency of Cre-mediated recombination and lineage-tracing of the fate of recombined acinar cells. *Table 1* summarizes the genotypes of mice used throughout this study; *Figure 2—figure supplement 1* schematically depicts the alleles in each genotype.

In initial studies, 6- to 8-week-old mice were administered tamoxifen (TM) at 0.17 mg/g body weight, and pancreata were harvested 9 months later (*Figure 2A*). Compared with control samples, *Ptf1a* cKO pancreata exhibited intermittent ADM throughout the pancreas (*Figure 2B,C*). Metaplastic 'ductules' of *Ptf1a* cKO expressed Cytokeratin-19 (CK19), similar to normal ducts of control; however, *Ptf1a* cKO ductules appeared more dilated than control ducts (*Figure 2F,G*). *Ptf1a* cKO ductules also expressed the duct cell-restricted transcription factor SOX9 (*Figure 2J,K*), indicating a shift from an acinar to a duct-like differentiation state (*Kopp et al., 2012*). However, these metaplastic ductules did not have the histological morphology of PanINs (*Figure 2C*), nor did they stain positively for the PanIN-specific markers Claudin-18 (CLDN18) by IHC (*Westmoreland et al., 2012*) (*Figure 2O*) or acidic mucins by Alcian Blue histochemistry (*Hingorani et al., 2003*; *Kopp et al., 2012*) (*Figure 2S*). Interestingly, ADM in *Ptf1a* cKO mice was associated with no or scant inflammatory infiltrates, and the surrounding areas did not stain positively with Sirius Red (*Figure 2W*), a histochemical stain that highlights fibrotic collagen matrix (*Neuschwander-Tetri et al., 2000*).

We next tested if inactivation of *Ptf1a* sensitized acinar cells to oncogenic KRAS-mediated transformation and PanIN initiation. While intermittent PanIN formation was observed in *Kras*[G12D] mice (*Figure 2D*), pancreata from *Ptf1a* cKO; *Kras*[G12D] mice were uniformly composed of extensively distributed PanINs embedded in fibrotic stroma, with almost no remaining normal acinar tissue (*Figure 2E*). PanINs in both *Kras*[G12D] and *Ptf1a* cKO; *Kras*[G12D] mice were positive for the duct marker Cytokeratin-19 (*Figure 2H,I*) and the duct-cell transcription factor SOX9 (*Figure 2L,M*), as well as the PanIN markers CLDN18 (*Figure 2P,Q*) and Alcian Blue acidic mucin staining (*Figure 2T,U*). Interestingly, only the *Ptf1a* cKO; *Kras*[G12D] pancreata exhibited abundant Sirius Red staining, indicating widespread fibrotic injury (*Figure 2X,Y*). Taken together, these data indicate that loss of *Ptf1a* sensitizes acinar cells to ADM and dramatically increases their susceptibility to oncogenic KRAS transformation and PDAC initiation.

## Loss of *Ptf1a* expression is a rate-limiting step for PanIN initiation

Given the severity and robustness of PanIN formation in *Ptf1a* cKO; *Kras*[G12D] mice 9 months after TM administration, we next determined if loss of *Ptf1a* had a more acute effect on acinar cell transformation. To address this issue, 6- to 8-week-old mice were administered TM (0.17 mg/g) and pancreata were harvested 2 or 6 weeks thereafter (*Figure 3A*). To ensure that Cre-mediated recombination rates were comparable between genotypes, we determined the percentage of acinar cells expressing the *R26R*[EYFP] reporter at 2 weeks post-TM administration. We found similar acinar recombination rates of 21–25% between genotypes (*Figure 2—figure supplement 1*). As the efficiency of Cre-mediated recombination can vary between different target loci (*Liu et al., 2013*), we additionally compared the extent and distribution of PTF1A ablation to that of *R26R*[EYFP] activation. 3 days after TM administration (0.17 mg/g), there was a ~20% decrease in the number of PTF1A+ cells

**Table 1**. Nomenclature of mouse mutants used in this study

| Short-hand notation | Ptf1a alleles | Kras allele | Reporter allele |
|---|---|---|---|
| Control | *Ptf1a*[CreERT/+] | – | *R26R*[EYFP/+] |
| *Ptf1a* cKO | *Ptf1a*[CreERT/lox] | – | *R26R*[EYFP/+] |
| *Kras*[G12D] | *Ptf1a*[CreERT/+] | *Kras*[LSL-G12D/+] | *R26R*[EYFP/+] |
| *Ptf1a* cKO; *Kras*[G12D] | *Ptf1a*[CreERT/lox] | *Kras*[LSL-G12D/+] | *R26R*[EYFP/+] |

cKO, conditional knock-out.

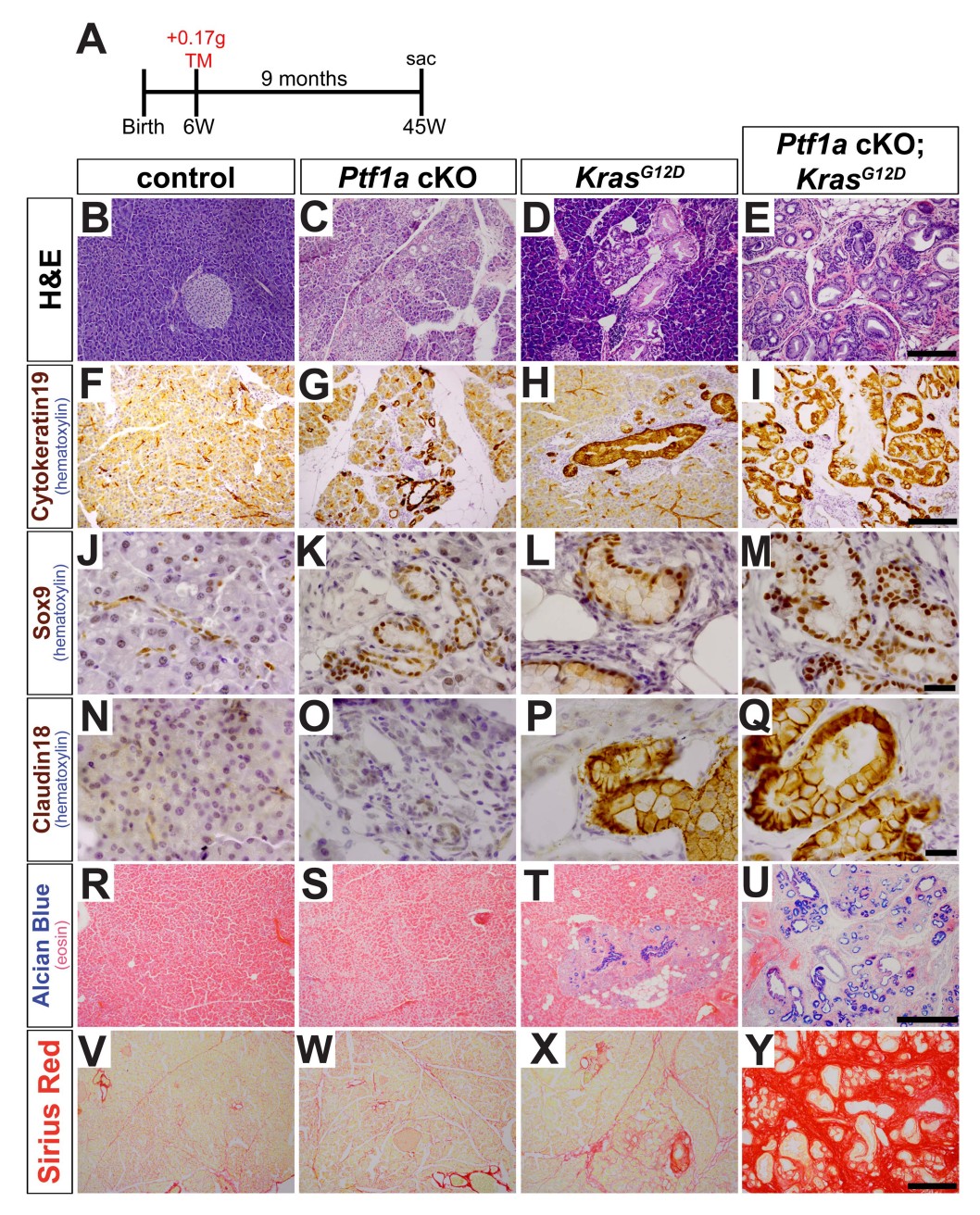

**Figure 2**. Loss of *Ptf1a* promotes acinar-to-ductal metaplasia and sensitizes acinar cells to KRAS-mediated transformation. (**A**) Mice of indicated genotypes were administered TM (0.17 mg/g) to induce recombination, and sacrificed 9 months later. (**B–E**) H&E staining of pancreata from mice of indicated genotypes. (**F–M**) IHC for the duct markers CK19 and SOX9, indicating upregulation in both acinar-to-ductal metaplasia (ADM) and PanINs. (**N–Q**) IHC for the PanIN marker, CLDN18, highlighting intermittent PanIN formation in $Kras^{G12D}$ mice and widespread lesion development in *Ptf1a* conditional knock-out (cKO); $Kras^{G12D}$. (**R–U**) Alcian Blue staining, indicating PanIN lesions in $Kras^{G12D}$ and *Ptf1a* cKO; $Kras^{G12D}$ pancreata. (**V–Y**) Sirius Red staining, highlighting local and widespread fibrosis in $Kras^{G12D}$ and *Ptf1a* cKO; $Kras^{G12D}$ mice, respectively. Scale bars: (**B–E**) 200 μm; (**F–I**) 200 μm; (**J–Q**) 25 μm; (**R–U**) 500 μm; (**V–Y**) 200 μm.

The following figure supplements are available for figure 2:

**Figure supplement 1**. Schematic of mouse alleles used in this study.

**Figure supplement 2**. *Ptf1a^CreERT^* deletion efficiency following tamoxifen treatment.

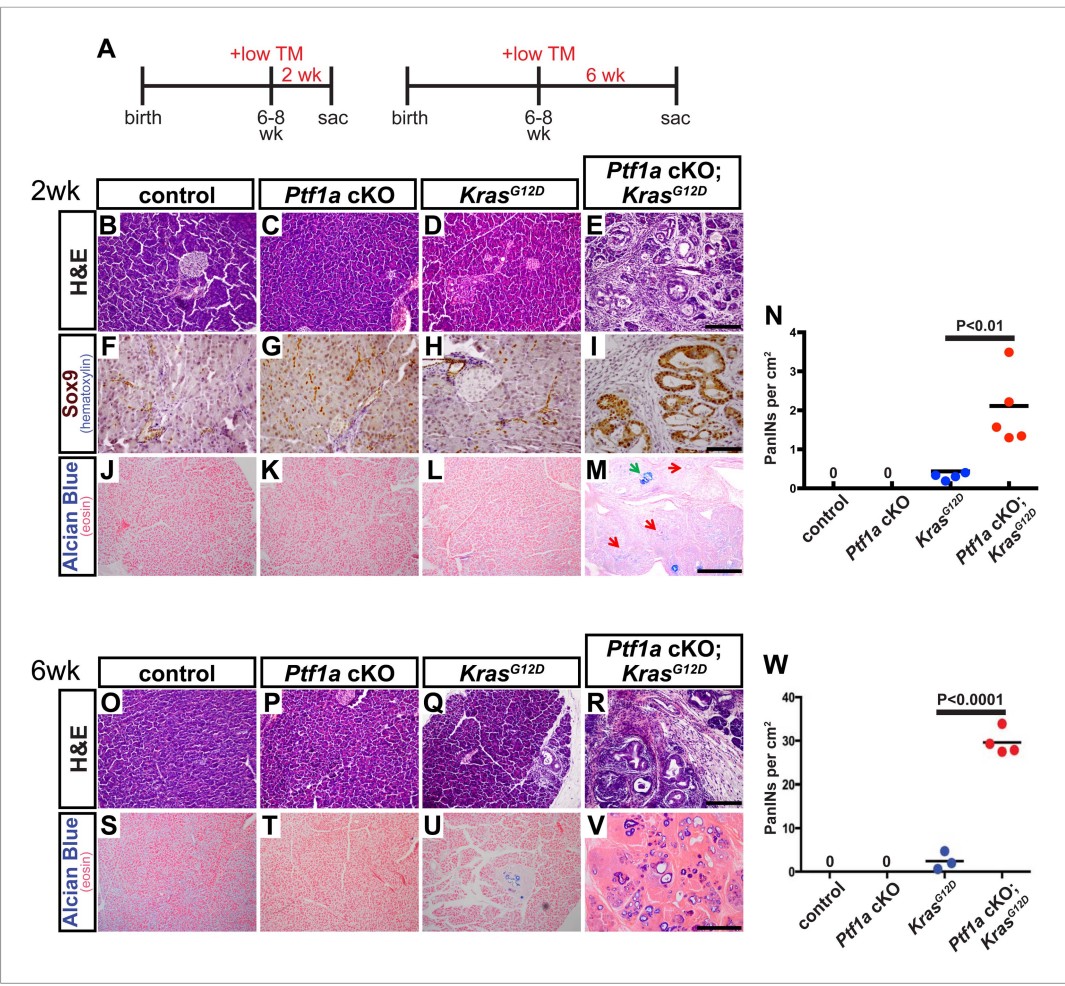

**Figure 3**. Loss of *Ptf1a* is a rate-limiting step in PanIN initiation. (**A**) Mice of specified genotypes were administered 0.17 mg/g body weight TM to induce Cre-mediated recombination and were sacrificed either 2 or 6 weeks later. (**B–E**) H&E staining of pancreata from mice of indicated genotypes 2 weeks after TM administration. (**F–I**) IHC for the ductal transcription factor SOX9, indicating upregulation in ADM and PanINs of *Ptf1a* cKO; *Kras*[G12D] pancreata. (**J–M**) Alcian blue staining, indicating PanIN lesions in *Ptf1a* cKO; *Kras*[G12D] pancreata. In panel (**M**), green arrow indicates an Alcian Blue+ lesion, while red arrows indicate ADM that is Alcian Blue-negative. (**N**) Quantification of the genotype-dependent PanIN burden; *Ptf1a* cKO; *Kras*[G12D] pancreata possessed significantly more PanINs at 2 weeks post-TM than *Kras*[G12D] mice (p < 0.01). (**O–R**) H&E staining of pancreata from mice of indicated genotypes 6 weeks after TM administration. (**S–V**) Alcian Blue staining, highlighting PanIN lesions in pancreata from *Kras*[G12D] mice and *Ptf1a* cKO; *Kras*[G12D] mice. (**W**) Quantification of PanINs at 6 weeks post-TM. *Ptf1a* cKO; *Kras*[G12D] pancreata had ~15-fold more Alcian Blue+ PanINs at this time point than *Kras*[G12D] (p < 0.0001). Scale bars: (**B–E**) 200 μm; (**F–I**) 100 μm; (**J–M**) 500 μm; (**O–R**) 200 μm; (**S–V**) 500 μm.

The following figure supplements are available for figure 3:

**Figure supplement 1**. Microenvironmental remodeling in *Ptf1a* cKO; *Kras*[G12D] pancreata.

**Figure supplement 2**. Acinar-ductal reprogramming in 3D culture.

detected by immunofluorescence (*Figure 2—figure supplement 2*). Importantly, the majority (~75%) of EYFP+ cells were PTF1A-negative at this dose of TM (*Figure 2—figure supplement 2*), indicating that activation of EYFP provides an approximate surrogate for deletion of *Ptf1a*.

Interestingly, this level of *Ptf1a* deletion alone did not produce ADM or other histologically detectable effects at 2 weeks post-TM, compared to control mice (*Figure 3B,C*). While *Kras*[G12D] pancreata exhibited few or no PanINs at this time point, there was widespread induction of ADM,

leukocyte infiltration, fibrosis, and PanIN initiation in *Ptf1a* cKO; *Kras*$^{G12D}$ pancreata (*Figure 3D,E* and *Figure 3—figure supplement 1*). We further confirmed that acinar-derived ADM and PanINs were being reprogrammed to a duct-like fate based on expression of the ductal transcription factor SOX9. While only normal ducts expressed SOX9 in control pancreata, PanINs and ADM in *Ptf1a* cKO; *Kras*$^{G12D}$ were SOX9$^+$ at 2 weeks post-TM (*Figure 3F–I*). These data are consistent with a recent study indicating that *Sox9* is necessary but not sufficient for the earliest stages of mouse PanIN initiation (*Kopp et al., 2012*).

To quantify lesion burden, we stained pancreata from all genotypes with Alcian Blue to highlight acidic mucin-rich PanINs (*Figure 3J–M*). Following a counting procedure established in our lab (*De La et al., 2008*), we observed a ~sixfold increase in the frequency of PanINs in *Ptf1a* cKO; *Kras*$^{G12D}$ mice compared to mice expressing *Kras*$^{G12D}$ alone (*Figure 3N*). This is likely an underestimation of overall phenotypic change, since ADM, which precedes PanIN formation, does not stain with Alcian Blue. Based on histological inspection, ADM is widespread in *Ptf1a* cKO; *Kras*$^{G12D}$ mice at 2 weeks post-TM, but negligible in *Kras*$^{G12D}$ pancreata (*Figure 3D,E,L,M*).

Initiation and progression of PDAC involves interactions between KRAS-active epithelial cells and their stromal microenvironment, with local inflammation being commonly associated with more rapid tumorigenesis (*Gukovsky et al., 2013*). We observed that PanINs developing after 2 weeks in *Ptf1a* cKO; *Kras*$^{G12D}$ mice, identified by CLDN18 staining, were consistently surrounded by CD45$^+$ leukocytes, indicating interactions between transformed epithelial cells and inflammatory cells (*Figure 3—figure supplement 1*). Because activation of pancreatic stellate cells is a hallmark of PDAC, we assessed the activation state of these cells using the marker α-smooth muscle actin (SMA). While SMA-positive cells were observed around blood vessels in pancreata of all genotypes, lobules of *Ptf1a* cKO; *Kras*$^{G12D}$ pancreata affected by ADM and PanIN initiation exhibited widespread SMA$^+$ fibroblasts surrounding ADM and PanIN lesions (*Figure 3—figure supplement 1*). Staining with the fibrosis marker Sirius Red confirmed that PanIN-associated fibroblasts of *Ptf1a* cKO; *Kras*$^{G12D}$ pancreata were actively secreting collagenous matrix (*Figure 3—figure supplement 1*), indicating activation of stellate cells only 2 weeks after Cre-mediated recombination. The activation of stellate cells and fibrotic phenotype observed in *Ptf1a* cKO; *Kras*$^{G12D}$ pancreata (*Figure 3—figure supplement 1*) is likely a reaction to the high level of acinar cell transformation rather than a direct reaction to *Ptf1a* deletion itself, as *Ptf1a* cKO pancreata with ADM do not stain with Sirius red (*Figure 2V*).

In order to determine the acinar cell-intrinsic consequences of *Ptf1a* deletion, we used a 3D culture system in which acini can undergo metaplasia into ductal cysts in response to mutant *Kras* or EGF receptor (EGFR) ligand stimulation, without the influence of other cell types (*Means et al., 2005*; *Ardito et al., 2012*). To induce widespread, acinar-specific *Kras* activation and/or *Ptf1a* deletion, we treated mice with three daily doses of tamoxifen at 0.25 mg/ml, a treatment paradigm that we found to drive widespread recombination (see below). Acinar cell clusters from control, *Ptf1a* cKO, *Kras*$^{G12D}$, and *Ptf1a* cKO; *Kras*$^{G12D}$ were isolated at 3 days after the final TM dose, prior to the appearance of any histological abnormalities, and embedded in a collagen matrix, as previously described (*Means et al., 2005*; *Ardito et al., 2012*). Neither control nor *Ptf1a* cKO acinar clusters underwent spontaneous cyst conversion, in the absence of added growth factors, implying that loss of *Ptf1a* is not sufficient for acinar-ductal reprogramming. As expected, acini of both genotypes generated CK19+ ductal cysts in response to the EGFR ligand TGFα (data not shown). By contrast, *Kras*$^{G12D}$ activation was sufficient for generation of acinar-derived cysts; importantly, *Ptf1a* cKO; *Kras*$^{G12D}$ acini formed significantly larger cysts than those derived from *Kras*$^{G12D}$ pancreata (*Figure 3—figure supplement 2*). These results are consistent with our in vivo data and suggest that acinar cell loss of *Ptf1a* enhances KRAS-mediated transformation independent of effects on the stromal microenvironment.

A generally similar synergy between *Kras*$^{G12D}$ and *Ptf1a* cKO was observed in vivo at the 6-week post-tamoxifen time point. *Ptf1a* cKO pancreata remained histologically unchanged compared to control, as at 2 weeks post-TM, while intermittent PanIN-1 lesions were observed in *Kras*$^{G12D}$ pancreata (*Figure 3O–Q*). *Ptf1a* cKO; *Kras*$^{G12D}$ pancreata, by contrast, were completely overrun by PanINs at this time point (*Figure 3R*), most of which stained positively with Alcian Blue (*Figure 3V*). Quantifying PanIN lesions by Alcian Blue staining, we observed a >15-fold increase in *Ptf1a* cKO; *Kras*$^{G12D}$ compared to mice expressing *Kras*$^{G12D}$ alone (*Figure 3W*). As we did not score more than one lesion per individual anatomic lobule, to avoid double-counting large or discontinuous lesions, this number likely underestimates the overall PanIN burden in *Ptf1a* cKO; *Kras*$^{G12D}$ pancreata given the

likelihood of multiple initiation events per lobule. Altogether, the dramatic acceleration of PanIN development upon *Ptf1a* deletion suggests that downregulation of this TF is a rate-limiting step for KRAS-driven pancreatic tumorigenesis.

## Extensive deletion of *Ptf1a* promotes rapid but incomplete acinar-ductal metaplasia

As we were surprised that a moderate level of acinar cell recombination (~25%) failed to produce an overt, short-term phenotype in *Ptf1a* cKO pancreata (*Figure 3*), we tested if more pervasive deletion of *Ptf1a* would produce a more robust reprogramming phenotype. Control and *Ptf1a* cKO mice were administered a higher dose of TM (0.25 mg/g) by oral gavage on three consecutive days (a net 4.5-fold higher dose than previously) and were harvested 2 weeks later (*Figure 4A*). Quantification of EYFP+ acinar cells following this TM regimen demonstrated a recombination frequency of ~65% (*Figure 4—figure supplement 1*). Additionally, we quantified the number of PTF1A-deficient acinar cells at 3 days after the final TM gavage, and found that only ~15% of all pancreatic cells retained nuclear PTF1A, compared with ~82% in TM-untreated controls (*Figure 2—figure supplement 2*). As with low-dose TM, described above, the majority (>90%) of EYFP+ cells were PTF1A-negative at 3 days post-TM, confirming that EYFP expression highlights acinar cells deleted for *Ptf1a* (*Figure 2—figure supplement 2*). The apparently greater extent of PTF1A ablation, relative to EYFP activation, may imply the existence of *Ptf1a*-deleted cells within the EYFP-negative population; such an observation would be consistent with previous evidence of locus-specific Cre deletion efficiencies (*Liu et al., 2013*).

2 weeks following high-dose TM, *Ptf1a* cKO pancreata were less than half the mass of their control counterparts (*Figure 4B*). Immunofluorescence revealed that while nearly all EYFP+ acinar cells expressed the acinar marker carboxypeptidase A1 (CPA1) in controls, this marker was lost from approximately 15% of EYFP+ cells in *Ptf1a* cKO tissues, indicating loss of the normal differentiation state (*Figure 4C–E*). Histologically, *Ptf1a* cKO pancreata exhibited extensive acinar disorganization and dilation as well as sporadic upregulation of CK19 within acinar structures, suggestive of early stages of ADM (*Figure 4F–I*). CK19+ acinar cells (defined by EYFP co-expression) were consistently surrounded by CD45+ leukocytes (*Figure 4—figure supplement 2A–C*), consistent with an intimate association between metaplasia and inflammatory cell recruitment (*Liou et al., 2013*; *Murtaugh and Keefe, 2015*). Nonetheless, *Ptf1a* cKO pancreata did not exhibit a general pancreatitis phenotype (*Figure 4—figure supplement 2A–C*) nor did they exhibit a detectable increase in epithelial cell apoptosis (*Figure 4—figure supplement 2D–F*). In addition, we found that treatment of wild-type mice with high-dose TM was not sufficient to induce pancreatic inflammation (*Figure 4—figure supplement 3*), suggesting that the stronger phenotype of high-dose *Ptf1a* cKO mice, relative to low-dose, was not due to stimulation of ADM by non-specific tissue damage.

Loss of PTF1A was accompanied by upregulation of SOX9 by the majority of EYFP+ cells, indicating partial reprogramming to a duct-like state (*Figure 4J–M*). Surprisingly, we also observed a significant (~fourfold) increase in the fraction of Ki67+ epithelial cells in *Ptf1a* cKO pancreata compared with control, suggesting that loss of PTF1A results in deregulation of proliferation as well as differentiation (*Figure 4—figure supplement 2G–I*). Taken together, these data indicate that *Ptf1a* is required to maintain acinar gene expression and quiescence, as well as prevent metaplasia to a duct-like state, potentially by inhibiting upregulation of SOX9.

## Loss of *Ptf1a* activates KRAS-dependency and fibroinflammatory pathways

In order to investigate further the mechanism of ADM after loss of *Ptf1a*, we performed RNA-seq on whole pancreata from three control and three *Ptf1a* cKO mice, each of which received three doses of TM (0.25 mg/g) to induce maximal recombination 2 weeks prior to RNA extraction. Initial analysis of RNA-seq data sets by *edgeR* (*Robinson et al., 2010*), setting a false discovery rate (FDR) threshold of 0.05, identified significant changes in expression of over 3000 total genes (*Figure 5A*). Consistent with our immunostaining (*Figure 4*), among the most significantly downregulated mRNAs were *Ptf1a* (18.4-fold) and *Cpa1* (5.45-fold), while *Sox9* was significantly upregulated (4.61-fold) in *Ptf1a* cKO pancreata (*Figure 5A*). Additional downregulated mRNAs

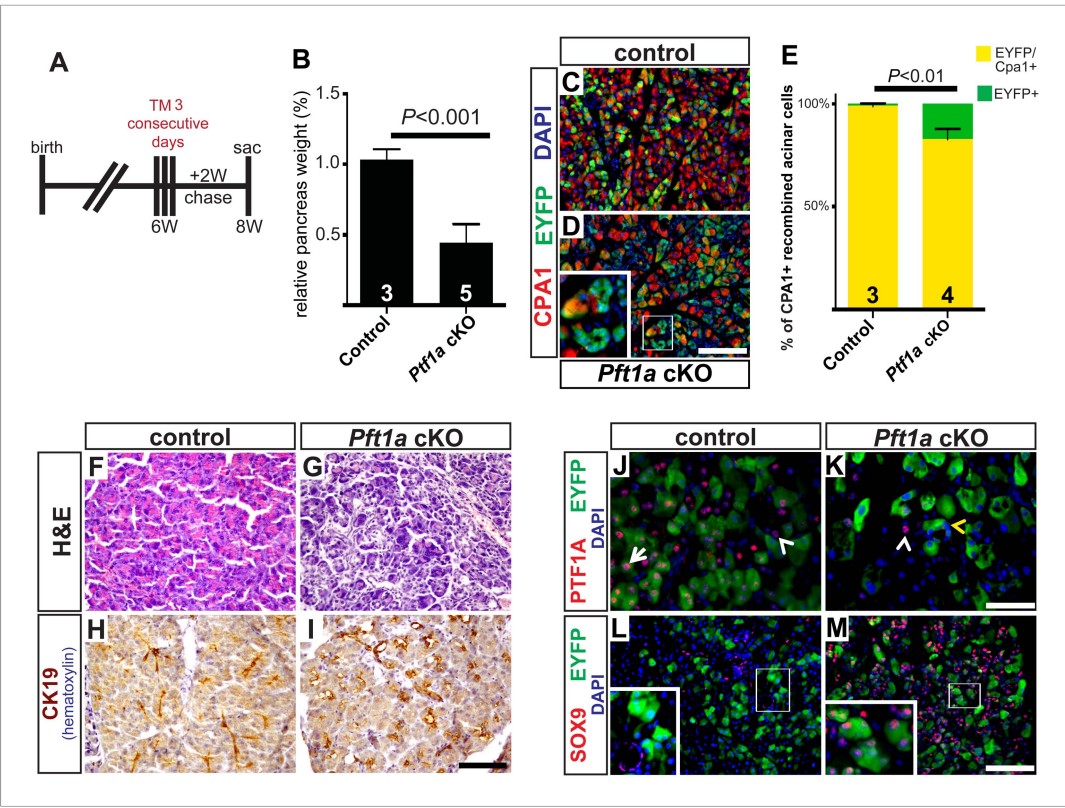

**Figure 4**. Widespread loss of *Ptf1a* promotes rapid acinar-to-ductal metaplasia. (**A**) Control and *Ptf1a* cKO mice were administered TM (0.25 mg/g) on three consecutive days and sacrificed following a 2-week chase period. (**B**) Pancreas mass, measured as a percent of body weight, was significantly decreased in *Ptf1a* cKO mice 2 weeks after TM administration. (**C**, **D**) Immunofluorescence for the acinar enzyme carboxypeptidase A1 (CPA1) (red) and Cre reporter *R26R$^{EYFP}$* (green). Nuclei are labeled with DAPI (blue). Inset highlights EYFP+, CPA1-negative acinar cells forming duct-like structures in *Ptf1a* null pancreata. (**E**) Quantification of CPA1 expression by EYFP+ (Cre-recombined) cells in control and *Ptf1a* cKO pancreata (control n = 3, *Ptf1a* cKO n = 4, p < 0.01). (**F**, **G**) H&E staining of control and *Ptf1a* cKO pancreata 2 weeks after high-dose TM administration. (**H**, **I**) IHC for the duct marker CK19 highlighting areas of ADM in *Ptf1a* cKO pancreata. (**J**, **K**) Immunofluorescence for PTF1A (red) and the Cre reporter *R26R$^{EYFP}$* (green). White arrow indicates an EYFP+ cell expressing PTF1A in control; white arrowheads indicate non-recombined PTF1A+ cells; yellow arrowhead indicates a recombined, PTF1A-negative cell undergoing metaplasia in *Ptf1a* cKO. (**L**, **M**) Immunofluorescence for the duct transcription factor SOX9 (red) and the Cre reporter *R26R$^{EYFP}$* (green). Insets highlight restricted expression of SOX9 in controls and upregulation of SOX9 within EYFP+ acinar cells of *Ptf1a* cKO. Scale bars: (**C**, **D**) 100 μm, (**F**–**I**) 200 μm, (**J**, **K**) 50 μm, (**L**, **M**) 100 μm.

The following figure supplements are available for figure 4:

**Figure supplement 1**. Cre-mediated recombination rates following high-dose tamoxifen treatment.

**Figure supplement 2**. Loss of Ptf1a promotes pancreatic epithelial transdifferentiation and proliferation.

**Figure supplement 3**. High-dose tamoxifen administration does not induce pancreatitis.

included a wide variety of digestive enzymes and other secreted proteins characteristic of the exocrine acinar phenotype, consistent with the long-standing hypothesis that they are directly regulated by PTF1A (*Rose et al., 2001*; MacDonald et al., in preparation).

Given our finding that loss of *Ptf1a* strongly potentiates KRAS-induced PanIN initiation (*Figures 2, 3*), we analyzed the expression of genes previously implicated in KRAS signaling and PDAC development. Interestingly, tumor suppressors classically associated with PDAC, such as *p53* (*Trp53*), *Cdkn2a/Ink4a*, *Pten*, *Brca2*, and *Smad4*, were not significantly downregulated in the absence of *Ptf1a* (data not shown),

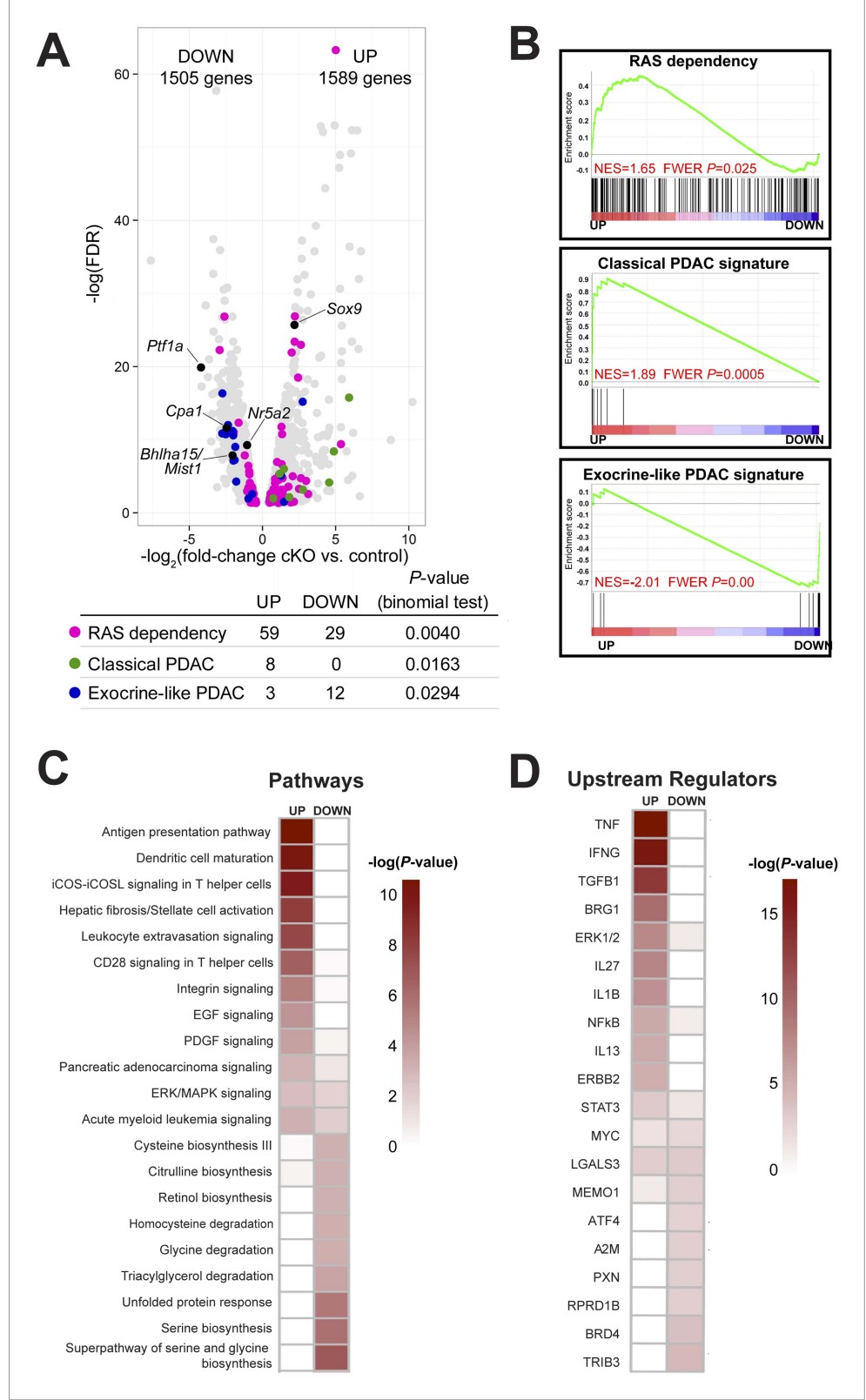

**Figure 5**. Ptf1a suppresses fibroinflammatory pathways and oncogenic KRAS associated gene signatures. (**A**) Volcano plot showing differentially expressed genes (false discovery rate [FDR] <0.05; gray) in *Ptf1a* cKO pancreata, relative to control. Individual genes are labeled and highlighted in black. Genes belonging to signatures characteristic of RAS dependency, classical and exocrine-like pancreatic ductal adenocarcinoma (PDAC) are

*Figure 5. Continued*

highlighted in pink, green, and blue, respectively. Table below indicates p-values from binomial test for enrichment of gene signatures within up- or down-regulated genes. (**B**) Gene Set Enrichment Analysis (GSEA) enrichment plots of differentially expressed genes between *Ptf1a* cKO and control indicating positive enrichment of RAS dependency and classical PDAC signatures and negative enrichment of exocrine-like PDAC signature genes. (**C**, **D**) Ingenuity Pathway Analysis (IPA, Qiagen Redwood City, www.qiagen.com/ingenuity) was used to identify differentially expressed pathways and upstream regulators in *Ptf1a* cKO pancreata. (**C**) Heat map of pathways that are significantly increased and decreased upon *Ptf1a* deletion. (**D**) Heat map of upstream pathways and regulators predicted to drive the observed changes in gene expression. Color scale is indicative of the -log p-value (significance). All analyses are based on a ±2.0-fold expression threshold. Full details of the data set and analyses can be found in the supplementary data files.

leading to the notion that the susceptibility of *Ptf1a* cKO pancreata to KRAS involves a novel mechanism distinct from canonical tumor suppression pathways. By contrast, we found that two acinar-specific transcription factors previously implicated in suppressing PanIN development, *Bhlha15* (commonly referred to as *Mist1*) and *Nr5a2* (*Shi et al., 2009b*; *Flandez et al., 2014*; *von Figura et al., 2014b*), were downregulated in *Ptf1a* cKO mice, consistent with PTF1A acting at or near the top of a regulatory hierarchy responsible for maintaining acinar identity and suppressing tumorigenesis (*Figure 5A*).

In human cell lines derived from pancreatic and other cancers, dependence on KRAS signaling correlates with expression of specific gene signatures, including genes whose activity is required to sustain RAS activity and malignancy (*Singh et al., 2009*; *Loboda et al., 2010*). We found that previously identified RAS dependence genes were significantly enriched, by binomial test, among mRNAs upregulated in *Ptf1a* cKO tissue (*Figure 5A*). Within this signature were some of the most highly upregulated mRNAs in our data set, such as *Tspan1* (32.4-fold increase), *Slc1a2* (62.2-fold increase), *Fut2* (41.4-fold increase), and *Egr1* (6.2-fold increase) (*Supplementary files 1, 2*). The preferential upregulation of RAS dependency genes in *Ptf1a* cKO was confirmed by Gene Set Enrichment Analysis (GSEA) (*Subramanian et al., 2005*) (*Figure 5B*), and suggests that loss of PTF1A results in a phenotypic shift toward a KRAS-permissive phenotype. RAS dependency is characteristic of human PDAC cell lines and primary tumors with a 'classical', duct-enriched gene expression profile (*Collisson et al., 2011*). We find that the classical PDAC signature is also preferentially upregulated upon *Ptf1a* deletion, while the distinct 'exocrine-like' PDAC signature, largely comprising acinar-specific secreted proteins, is downregulated (*Figure 5A,B*). These results therefore strongly suggest not only that PTF1A maintains acinar differentiation, including expression of genes marking an acinar-like subset of human PDAC, but also that PTF1A suppresses an alternative gene expression program that facilitates KRAS signaling activity.

To identify biological pathways that were activated or attenuated by *Ptf1a* deletion, we analyzed this RNA-seq data set using Qiagen's Ingenuity Pathway Analysis (IPA, QIAGEN Redwood City, www.ingenuity.com) (*Thomas and Bonchev, 2010*; *Kramer et al., 2014*). We analyzed canonical pathways using three different thresholds of gene expression (1.5-fold up/downregulation, 2.0-fold up/downregulation, and 3.0-fold up/downregulation; *Supplementary files 3–5*). At an upregulation threshold of 2.0, deletion of *Ptf1a* significantly affected over 300 pathways, several of which have an established role in PDAC initiation. These included T-helper cell-signaling pathways (*McAllister et al., 2014*), stellate-cell activation and fibrosis (*Sherman et al., 2014*), and epidermal growth factor (EGF) signaling (*Ardito et al., 2012*; *Navas et al., 2012*) (*Figure 5C*). A general 'pancreatic adenocarcinoma signaling' pathway was also upregulated, consisting primarily of genes involved in PI-3-kinase and JAK/STAT signaling. We also used IPA Upstream Regulator Analysis to predict upstream signaling mediators that could explain the changes in gene expression within our data set (*Kramer et al., 2014*). The predicted upregulated mediators were consistent across multiple expression thresholds and included TNF-α, TGF-β, IL-1β, NFκB, and the SWI/SNF component Smarca4/Brg1 (*Figure 5D*). All of these signaling pathways have been implicated in PDAC initiation and progression (*Bardeesy et al., 2006*; *Adrian et al., 2009*; *Khasawneh et al., 2009*; *Maniati et al., 2011*; *Daniluk et al., 2012*; *Maier et al., 2013*; *Gore et al., 2014*; *von Figura et al., 2014a*). Thus, we propose that loss of *Ptf1a* alters cell state at multiple levels, ultimately promoting gene expression and signaling activities that are supportive of KRAS transformation.

## Caerulein-induced pancreatitis is sufficient to reprogram *Ptf1a*-deficient acinar cells

Among the upstream mediators activated in the *Ptf1a* cKO model are TNF-α and NFkB, both of which promote ADM and inflammation in pancreatitis and amplify KRAS activity in pancreatic tumorigenesis (*Maniati et al., 2011*; *Daniluk et al., 2012*; *Huang et al., 2013*; *Maier et al., 2013*; *Sendler et al., 2013*). As *Ptf1a* deletion upregulates other pathways characteristic of pancreatic injury, such as stellate-cell activation, TGF-β signaling, and dendritic cell maturation (*Bedrosian et al., 2011*; *Erkan et al., 2012*), we were interested to determine if loss of *Ptf1a* would sensitize acinar cells to injury-induced reprogramming even without oncogenic *KRAS*.

To test this hypothesis in vivo, we deleted *Ptf1a* via high-dose TM administration (three doses of 0.17 mg/g), which induced a recombination rate of ~65% (*Figure 4—figure supplement 1*). At 1 week post-TM, acute pancreatitis was induced by two consecutive days of treatment with the secretagogue caerulein, as previously described (*Jensen et al., 2005*; *Keefe et al., 2012*), and pancreata were harvested 1 week later (*Figure 6A*). As a control for caerulein injections, additional TM-treated *Ptf1a* cKO and control mice were administered saline vehicle alone. As previously reported, control mice recovered from caerulein treatment and were indistinguishable from saline-injected controls after 1 week (*Figure 6B–D*). In contrast, *Ptf1a* cKO mice subjected to caerulein-induced pancreatitis exhibited widespread acinar atrophy, persistent inflammation, fibrotic stroma, and the appearance of mucinous metaplastic structures (*Figure 6E,F*). These abnormal ductules were Alcian Blue-reactive, similar to PanINs (*Figure 6G*), although staining for the PanIN-specific markers CLDN18 and MUC5AC was observed in only rare and isolated lesions (*Figure 6—figure supplement 1A,B*). Consistent with the overall distorted histology (*Figure 6E*) and atrophy (*Figure 6H*) of caerulein-treated *Ptf1a* cKO mice, no normal amylase+ acinar clusters could be detected in these pancreata, in contrast to controls (*Figure 6I–L*). Acinar-derived EYFP+ cells in caerulein-treated *Ptf1a* cKO pancreata were instead integrated within CK19+ duct-like structures, suggesting that pancreatitis synergizes with loss of *Ptf1a* to cause a rapid loss of acinar gene expression and complete reprogramming to a duct-like fate (*Figure 6I–L*).

As our findings in *Ptf1a* cKO; *Kras*^G12D mice indicate that loss of PTF1A enhances the transforming activity of mutant KRAS, we were interested to determine if development of mucinous metaplasia involved enhanced signaling through endogenous RAS. The MEK-ERK pathway is a major regulator of KRAS-induced acinar reprogramming (*Collins et al., 2014*), and we found that nearly all metaplastic lesions of caerulein-treated *Ptf1a* cKO mice exhibited robust nuclear phospho-ERK staining (*Figure 6—figure supplement 1C–F*). Phospho-ERK was undetectable in saline-treated *Ptf1a* cKO mice, or control mice under either treatment. Taken together, these data demonstrate that PTF1A is necessary for acinar-cell redifferentiation and resolution of tissue injury following acute pancreatitis. In the absence of PTF1A, a persistent inflamed microenvironment may have tumor promoter-like activity, enhancing KRAS-MEK-ERK signaling to induce transformation (*Gukovsky et al., 2013*; *Murtaugh, 2014*).

## *Ptf1a* heterozygosity promotes PDAC by increasing the frequency of initiating events

The above studies rely on genetic deletion of *Ptf1a*, a process without clear parallel in human disease: somatic mutations of *PTF1A* are not observed in human PDAC, according to the Catalogue of Somatic Mutations in Cancer (COSMIC) database (cancer.sanger.ac.uk). *PTF1A* is more likely to be downregulated by an epigenetic mechanism, for example, via attenuation of the positive autoregulatory loop by which PTF1A maintains its own expression and that of its partner transcription factors (*Masui et al., 2008*). Impaired expression of PTF1-network components, lowering the threshold for KRAS-mediate reprogramming and transformation, might explain the dosage-sensitive requirement for *Nr5a2* in preventing PanIN formation (*Flandez et al., 2014*; *von Figura et al., 2014b*). To determine if the role of *Ptf1a* itself is dosage-sensitive, we generated mice of the 'KC' genotype, using the *Pdx1-Cre* driver to activate *Kras*^LSL-G12D throughout the pancreas (*Aguirre et al., 2003*; *Hingorani et al., 2003*; *Murtaugh, 2014*), and which were either heterozygous for a germ line deletion of *Ptf1a* (*Pdx1-Cre; Kras*^LSL-G12D; *Ptf1a*^Δ/+) or remained homozygous *Ptf1a* wild type. We harvested pancreata at 1 month of age, at which time PanIN formation is usually minimal in KC mice, and quantified PanIN burden by Alcian Blue staining. Mice heterozygous for *Ptf1a* had increased

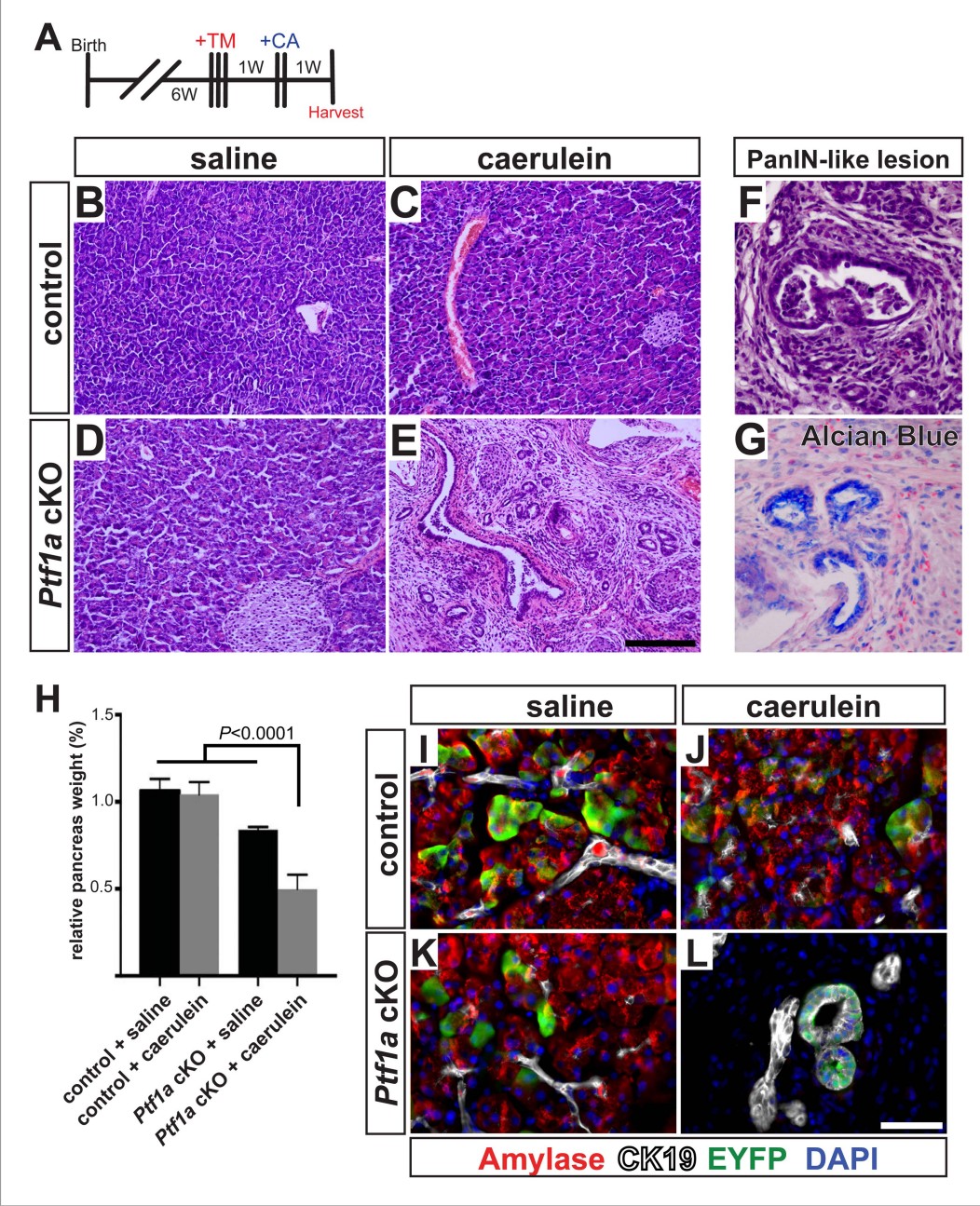

**Figure 6**. *Ptf1a* is necessary for acinar cell regeneration and suppression of dysplasia following induced pancreatitis. (**A**) 6- to 8-week-old control and *Ptf1a* cKO mice were administered three doses of TM (0.17 mg/g) on consecutive days. 1 week later, mice were administered eight hourly injections of caerulein or saline vehicle, on two consecutive days. Mice were sacrificed 1 week following caerulein treatment. (**B**–**E**) H&E staining on control and *Ptf1a* cKO pancreata (n = 4–5 per group) 1 week following caerulein treatment. (**F**) H&E stain highlighting a PanIN-like lesion in caerulein-treated *Ptf1a* cKO. (**G**) Alcian Blue-positive lesions in caerulein-treated *Ptf1a* cKO. (**H**) Relative pancreas size, measured as a percent of body weight, among treatment groups (n = 4–5 per group, p < 0.01). (**I**–**L**) Immunofluorescence for amylase (red), CK19 (white), and the Cre reporter *R26R^EYFP* (green), in pancreata of control and *Ptf1a* cKO treated with saline or caerulein. EYFP+ cells of caerulein-treated cKO have downregulated amylase and contribute to CK19+ PanIN-like structures. Scale bars: (**B**–**E**) 200 μm, (**I**–**L**) 50 μm.

The following figure supplement is available for figure 6:

**Figure supplement 1**. Mucinous metaplasia associated with hyperactive MEK-ERK signaling in caerulein-treated Ptf1a cKO pancreata.

PanINs at this early stage, compared to *Ptf1a*<sup>+/+</sup> littermates (*Figure 7A–C*). This result is consistent with a dosage-sensitive function for PTF1A, such that reduced levels or activity already begin to destabilize acinar differentiation in the face of oncogenic insults.

In humans, increased PanIN burden in early life is associated with familial risk of PDAC, suggesting that mutations driving genetic predisposition to PDAC act at the level of tumor initiation (*Brune et al.,*

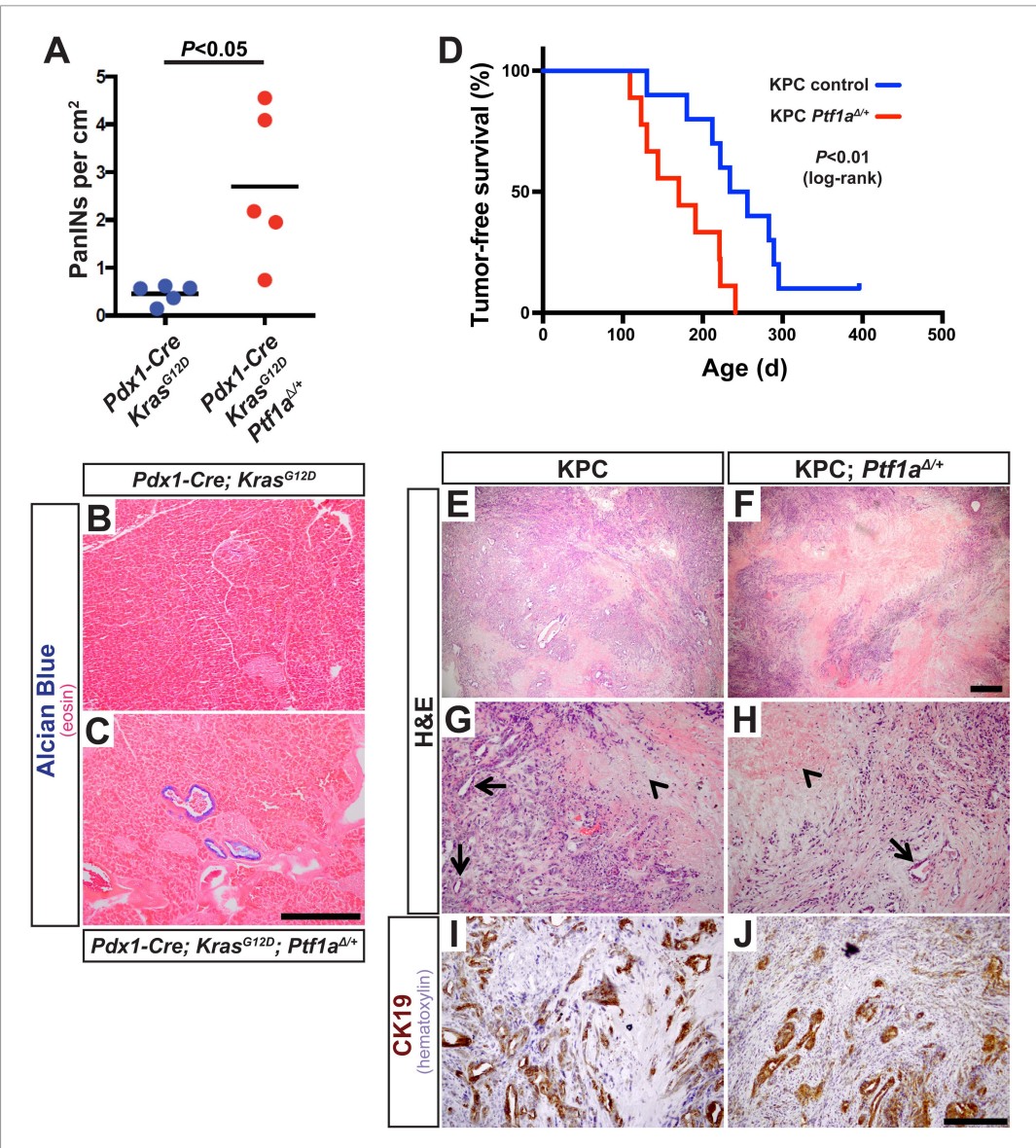

**Figure 7**. *Ptf1a* heterozygosity increases the frequency of PanINs and allows for rapid progression of PDAC. (**A**) Quantification of PanINs in pancreata from 1-month-old *Pdx1-Cre; Kras*<sup>G12D</sup> and *Pdx1-Cre; Kras*<sup>G12D</sup>; *Ptf1a*<sup>Δ/+</sup> mice (n = 5 per genotype, p < 0.05). (**B**, **C**) Representative Alcian Blue and Eosin staining from 1-month-old mice of indicated genotypes. (**D**) Kaplan–Meier analysis from KPC mice (*Pdx1-Cre; Kras*<sup>G12D</sup>; *p53*<sup>lox/+</sup>; *Ptf1a*<sup>+/+</sup>, blue line) and KPC; *Ptf1a*<sup>Δ/+</sup> mice (red line) (Log-Rank test p < 0.01). (**E–H**) H&E staining on tumors from both KPC and KPC:*Ptf1a*<sup>Δ/+</sup> mice at low and high magnification. (**G**, **H**) Arrows indicate ductile epithelial cells and arrowheads indicate areas of necrosis. (**I**, **J**) IHC for CK19 on tumor specimens from mice of indicated genotypes. Scale Bars: (**B**, **C**) 500 μm, (**E**, **F**) 500 μm, (**G–J**) 200 μm.

The following figure supplement is available for figure 7:

**Figure supplement 1**. Liver metastases in KPC mice heterozygous for *Ptf1a*.

*2006*; *Shi et al., 2009a*). We therefore hypothesized that decreased *Ptf1a* dosage would promote cancer susceptibility by increasing the rate of PanIN initiation. Therefore, we utilized the well-characterized 'KPC' model of mouse PDAC in which heterozygous loss of *p53* (official gene symbol *Trp53*) is added to the *Pdx1-Cre; Kras^{LSL-G12D}* genotype (*Hingorani et al., 2005*; *Rhim et al., 2012*). As above, KPC mice (*Pdx1-Cre; Kras^{LSL-G12D}; p53^{lox/+}*) were generated on either *Ptf1a^{+/+}* or *Ptf1a^{Δ/+}* backgrounds, and animals were monitored for tumor-free survival. The results of Kaplan–Meier analysis showed that *Ptf1a*-heterozygous KPC mice developed PDAC much earlier than *Ptf1a^{+/+}* counterparts (*Figure 7D*, Log-rank test, p < 0.01). We observed prominent metastases to the liver in 3/9 *Ptf1a^{Δ/+}* KPC mice, but none in *Ptf1a^{+/+}* KPC controls (*Figure 7—figure supplement 1*). Importantly, despite the earlier onset of PDAC in KPC mice with *Ptf1a* heterozygosity, once tumors arose they were histologically indistinguishable between genotypes (*Figure 7E–H*). They contained classical features of human PDAC, including abundant fibrotic stroma surrounding CK19+ epithelial cells (*Figure 7I,J*) and substantial areas of necrosis. We therefore conclude that decreased *Ptf1a* gene dosage sensitizes pancreata to early KRAS-mediated PanIN initiation and rapid progression to PDAC.

## Discussion

Previously, we and others established that acinar-to-ductal reprogramming is a necessary step in PanIN initiation (*De La et al., 2008*; *Habbe et al., 2008*; *Kopp et al., 2012*). Several recent studies extended these findings, demonstrating that several genes required for PanIN and PDAC development appear to act at the level of acinar cell reprogramming (*Heid et al., 2011*; *Ardito et al., 2012*; *Kopp et al., 2012*; *Baer et al., 2014*; *Wu et al., 2014*; *Zhang et al., 2014*). Here, we demonstrate that the loss of a principal regulator of acinar cell identity, PTF1A, is sufficient to prompt rapid and extensive acinar-to-ductal metaplasia even in the absence of other exocrine insults (*Figure 4*). Additionally, we demonstrate that *Ptf1a*-deficient acinar cells are extremely sensitive to oncogenic transformation, as they undergo rapid and robust KRAS-mediated PanIN formation (*Figures 2, 3*).

Deletion of *Ptf1a* alone at moderate frequency (∼25%) did not produce detectable histological changes in the pancreas over the course of 2–6 weeks (*Figure 3*). By contrast, we observed rapid de-differentiation of *Ptf1a cKO* acinar cells generated under a high-TM dose regimen that produced >65% deletion (*Figure 4*). This is an important finding regarding potentially non-cell autonomous protective mechanisms working to offset PanIN/PDAC initiation. We propose two linked hypotheses: first, when *Ptf1a* is lost from individual cells, other acinar-specific transcription factors that prevent reprogramming and co-regulate PTF1 target genes, such as NR5A2 and BHLHA15/MIST1, are sufficient to maintain a differentiated phenotype over the short term. Second, we propose that the persistent differentiation of *Ptf1a cKO* acinar cells is promoted by interactions with neighboring *Ptf1a* WT cells, producing a phenomenon similar to the 'community effect' in embryonic development (*Gurdon et al., 1993*). However, with increasing TM-driven deletion, the fraction of *Ptf1a* WT acinar cells passes a tipping point, the community effect cannot be sustained, and ductal metaplasia is correspondingly rapid. At a molecular level, the protective effect of wild-type acinar cells could be mediated by their ability to dampen local inflammation, suggested by the upregulation of fibroinflammatory pathways in our RNA-seq analyses (*Figure 5*). This is also suggested by our finding that *Ptf1a* cKO pancreata exhibit sustained inflammation after acute injury, including the conversion of acinar cells to PanIN-like, Alcian Blue+ ductule structures (*Figure 6*). These resemble tubular complexes observed in human and mouse chronic pancreatitis (*Bockman et al., 1982*; *Strobel et al., 2007*), suggesting that dysregulation of PTF1A expression or function might be involved in the etiology of this disease and its well-known contribution to PDAC risk. We additionally demonstrate that inflammation and loss of *Ptf1a* synergize to drive sustained activation of the MEK-ERK pathway, a major effector of oncogenic and endogenous KRAS. Going forward, it will interesting to test whether MEK inhibitors are able to prevent acinar cell reprogramming in the context of chronic pancreatitis and/or decrease the risk of chronic pancreatitis progressing to PDAC.

Given the dramatic effects of *Ptf1a* deletion on transformation and inflammation, it will be important to determine which genes in our RNA-seq data set are directly suppressed or activated by PTF1A. It has been previously established that PTF1A regulates a network of transcription factors controlling acinar-specific gene expression (*Masui et al., 2008, 2010*). Among these are *Bhlha15/Mist1* and *Nr5a2*, both downregulated in the *Ptf1a cKO* condition (*Figure 5*), and both previously shown to inhibit PanIN development (*Shi et al., 2009b*; *Flandez et al., 2014*; *von Figura et al., 2014b*). Of these three genes, only *Ptf1a* is indispensable for acinar cell differentiation (*Krapp et al., 1998*;

*Pin et al., 2001*; *Kawaguchi et al., 2002*; *Holmstrom et al., 2011*; *von Figura et al., 2014b*), and it will be of interest to determine the relative rank of these factors as suppressors of cancer initiation and progression, and their epistatic relationship. It will also be useful to understand how KRAS, together with inflammatory and other insults, is capable of downregulating the expression and/or function of PTF1-network components during tumor initiation. Of note, recent studies indicate that oncogenic KRAS induces specific pathways dedicated to silencing tumor suppressor genes (*Wajapeyee et al., 2013*; *Serra et al., 2014*); a similar process may drive downregulation of *Ptf1a* and its partners during acinar cell reprogramming.

Because loss of *Ptf1a* strongly potentiated KRAS-mediated transformation (*Figures 2, 3*), we hypothesized that PTF1A inhibits KRAS-signaling activity in some capacity. Here, we demonstrate that loss of *Ptf1a* leads to upregulation of genes associated with KRAS-dependency in human cancer cells (*Singh et al., 2009*; *Loboda et al., 2010*). Future investigations should therefore move forward to test if different subtypes of human PDAC exhibit different extents of *PTF1A* repression, and whether variation in *PTF1A* expression within human PDAC correlates with KRAS-dependency or disease prognosis. Recent studies have classified ~1/3 of pancreatic cancers as 'exocrine-like', and several genes that are under *Ptf1a* control contribute to this signature (*Figure 5A,B*) (*Collisson et al., 2011*). Unfortunately, human *PTF1A* was not present on the microarray used in that study; nonetheless, their data suggest that PTF1A and its transcriptional targets are retained at low levels in some, but not all, cases of PDAC. Consistent with these previous reports, we found that sparse epithelial cells in human PanIN lesions retain nuclear PTF1A (*Figure 1—figure supplement 1*). In addition to supporting the contention that human PanINs and PDAC arise from mature acinar cells, these findings suggest that low levels of persistent PTF1A, held in check by epigenetic rather than genetic mechanisms, may be available for therapeutically targeted restoration. The fact that removal of a single allele of *Ptf1a* accelerates mouse PDAC development (*Figure 7D*) suggests that even incomplete inhibition of human PTF1A could promote acinar transformation and subsequent tumorigenesis. Our results suggest that PTF1A restoration provides an indirect approach to target KRAS-dependency in pancreatic cancer, inhibiting this currently 'undruggable', although ubiquitous, cancer-driving mutation (*Pasca di Magliano and Logsdon, 2013*). Additionally, our data suggest that PTF1A restoration may reduce inflammatory pathways that feed forward to synergize with oncogenic KRAS (*Maniati et al., 2011*; *Daniluk et al., 2012*; *Maier et al., 2013*). Future studies will focus on genetic PTF1A gain-of-function approaches to determine if sustained PTF1A expression can prevent and/or reverse acinar-to-ductal reprogramming, PanIN initiation and PDAC progression.

In summary, we show that acinar cell differentiation, maintained through PTF1A, suppresses multiple oncogenic pathways associated with PDAC initiation and progression. Our data suggest that PTF1A functions as a nodal point in PDAC initiation by maintaining acinar-cell gene expression, suppressing KRAS function, and resisting inflammation. The antagonism between KRAS and the pro-acinar transcription factor network captures, at a genetic level, the tension between differentiation and malignant transformation that has long been hypothesized to exist in cancer (*Harris, 1990*). Loss of normal differentiation and reprogramming of cell fate appear to occur during initiation of a diverse array of tumor types (*Blanpain, 2013*). Our results, for the first time, demonstrate that this process is rate-limiting for cancer development, thus, constituting a novel mechanism of tumor suppression. The mouse PanIN-PDAC model provides a new experimental system to relate genetic changes in cancer, such as *KRAS* mutation, to epigenetic changes such as PTF1A downregulation. Furthermore, understanding how PTF1A function is subverted during pancreatic cancer initiation, and whether its reactivation could suppress or reverse tumor development, may yield novel approaches to prevention and treatment.

## Materials and methods

### Mice

Experimental mice of the following genotypes have been previously described: $Ptf1a^{CreERT}$ ($Ptf1a^{tm2(cre/ESR1)Cvw}$ [*Kopinke et al., 2012*; *Pan et al., 2013*]), Pdx1-Cre ($Tg(Pdx1-cre)89.1Dam$ [*Gu et al., 2002*]), $Kras^{LSL-G12D}$ ($Kras^{tm4Tyj}$ [*Hingorani et al., 2003*]), $p53^{lox}$ ($Trp53^{tm1Brn}$ [*Marino et al., 2000*]), and $R26R^{EYFP}$ ($Gt(ROSA)26Sor^{tm1(EYFP)Cos}$ [*Srinivas et al., 2001*]). The $Ptf1a^{lox}$ allele ($Ptf1a^{tm3Cvw}$) was generated using homologous recombination in mouse ES cells at the Vanderbilt Transgenic Mouse/ Embryonic Stem Cell Shared Resource. The 5′ and 3′ loxP sites were placed 1.7 kb upstream and 2 kb downstream of the *Ptf1a* transcriptional start site, respectively. Full details will be provided elsewhere (Wright et al., in preparation). Mice with a germ line deletion allele of *Ptf1a*, $Ptf1a^\Delta$ were generated by

crossing *Ptf1a^lox* to the ubiquitous early deletor line *Sox2-Cre* (*Tg(Sox2-cre)1Amc* [*Hayashi et al., 2003*]). To activate CreERT-mediated recombination, mice were administered tamoxifen (Sigma, St. Louis, MO) dissolved in corn oil, via oral gavage at doses indicated in the text. All mouse experiments were carried out according to institutional and NIH guidelines.

## Human histological specimens

All human pathological specimens were de-identified before their use. The utilization of these human specimens is therefore not considered human subject research under the US Department of Human and Health Services regulations and related guidance (45 CDR Part 46). Paraffin embedded specimens were sectioned (6 µm) and IHC was performed for PTF1A, as described below. Samples were analyzed by NMK, MPB, and LCM.

## Tissue processing and histology

After euthanasia, pancreata were dissected in ice-cold phosphate-buffered saline solution (PBS), separated into multiple fragments, and processed for both frozen and paraffin sections as previously described (*De La et al., 2008*; *Keefe et al., 2012*; *Kopinke et al., 2012*). Briefly, tissues were fixed for paraffin embedding in zinc-buffered formalin (Z-FIX; Anatech, Battle Creek, MI), room temperature overnight, or 4% paraformaldehyde/PBS, 4°C 1–2 hr, followed by processing into Paraplast Plus (McCormick Scientific) or Tissue-Tek O.C.T. compound (Sakura Finetek, Torrance, CA). Paraffin and frozen sections were cut at thickness of 6 µm and 8 µm, respectively, and collected sequentially across multiples slides, with ∼100-µm spacing between individual sections on a single slide.

IHC and immunofluorescence followed our established procedures (*De La et al., 2008*; *Keefe et al., 2012*; *Kopinke et al., 2012*), including high-temperature antigen retrieval (Vector Unmasking Solution; Vector Laboratories, Burlingame, CA) prior to staining paraffin sections. Primary antibodies utilized in this study are listed in *Table 2*. Secondary antibodies, raised in donkey (Jackson Immunoresearch, West Grove, PA), were used at 1:250 dilution. Vectastain reagents and diaminobenzidine (DAB) substrate (Vector Laboratories) were used for IHC. Slides stained by immunofluorescence were counterstained with DAPI and mounted in Fluoromount-G (Southern Biotech), and photographed on an Olympus IX71 microscope, using MicroSuite software (Olympus America, Waltham, MA). Images were processed in Adobe Photoshop, with exposure times and adjustments identical between genotypes and treatment groups.

For Alcian Blue staining, paraffin sections were washed 5 min in 3% acetic acid, followed by a 10–12 min incubation in staining solution (1% Alcian Blue in 3% acetic acid), and extensive washing in 3% acetic acid. Sirius Red staining was performed on frozen sections that were fixed for 1 hr in Bouin's fixative at 55°C. Specimens were subsequently washed in dH$_2$O and stained for 1 hr in Picro-Sirius Red (American MasterTech, Lodi, CA). Following staining, specimens were rinsed in 0.5% acetic acid, dehydrated and equilibrated into xylene, and mounted with Permount.

## PanIN scoring

The entire tissue area of Alcian Blue/eosin-stained slides was photographed, at 4× original magnification, followed by photomerging (Adobe Photoshop) and surface area measurement using ImageJ software (NIH). Alcian Blue+ PanIN lesions were counted manually under the microscope, and PanIN burden calculated as the total number of Alcian Blue+ lesions per cm$^2$ surface area. As described in the text, metaplastic lesions that did not stain with Alcian Blue were not counted in the quantification. To avoid double-counting of potentially large and tortuous lesions, no more than one lesion was scored within an anatomically distinct pancreatic lobule (*De La et al., 2008*).

## 3D pancreatic acinar cultures

Acinar cultures were established according to previous publications (*Kurup and Bhonde, 2002*; *Means et al., 2005*; *Ardito et al., 2012*). Briefly, dorsal pancreata were minced in Hank's buffered saline solution and digested sequentially in 0.02% trypsin (5 min, 37°C) and 1 mg/ml collagenase P (Roche Applied Science, Mannheim, Germany; 15 min, 37°C), following filtration to eliminate undigested material, and repeated washing to eliminate debris and dead cells, acinar cell clusters were embedded in rat tail collagen gels (Corning, Corning, NY), and cultured in Waymouth's medium (Life Technologies, Carlsbad, CA) supplemented with 1% fetal bovine serum, 0.4 mg/ml soybean trypsin inhibitor, and 1 µg/ml dexamethasone. Cultures were fixed and imaged after 5 days. To

**Table 2**. Primary antibodies used in this study

| Antigen | Species | Source | Catalog # | Dilution |
|---|---|---|---|---|
| Amylase | Sheep | BioGenesis | 0480-0104 | 1:1000 |
| Cleaved-caspase-3 | Rabbit | Abcam | AB2302 | 1:1000 |
| Cd45 | Rat | eBioScience | 14-0451-82 | 1:2000 |
| Claudin-18 | Rabbit | Invitrogen | 700178 | 1:2000 |
| Cpa1 | Goat | R&D Systems | AF2765 | 1:1000 |
| Cytokeratin-19 | Rat | Developmental Studies Hybridoma Bank | – | 1:50 |
| Cytokeratin-19 | Rabbit | Abcam | AB133496 | 1:5000 |
| GFP | Chicken | Aves Labs Inc. | GFP-1010 | 1:5000 |
| Ki67 | Mouse | BD Biosciences | 550609 | 1:500 |
| Muc5ac | Mouse | NeoMarkers | 45M1 | 1:500 |
| Ptf1a | Rabbit | Chris Wright, Vanderbilt University | – | 1:5000 |
| Ptf1a | Goat | Chris Wright, Vanderbilt University | – | 1:5000 |
| Phospho-ERK1/2 (T202/Y204) | Rabbit | Cell Signaling | 9101 | 1:1000 |
| Sox9 | Rabbit | Millipore | AB5535 | 1:1000 |
| α-SMA | Rabbit | Abcam | AB32575 | 1:2000 |

SMA, smooth muscle actin.

quantify cyst size, we randomly selected >10 fields per mouse, imaged, and quantified the maximal diameter of each transformed cyst using ImageJ.

## Quantification of immunofluorescence images

In order to quantify $R26R^{EYFP}$ labeling, 10–12 randomly selected 20× fields per specimen (taken across multiple sections) were photographed. Using ImageJ software (NIH), cells co-expressing EYFP with the acinar differentiation markers Amylase or CPA1 were detected by additive image overlay of their staining with DAPI and anti-GFP, and counted using the Analyze Particles function, as described previously (*Keefe et al., 2012*; *Kopinke et al., 2012*). To ensure counting accuracy, random images were spot checked by manual counting using Adobe Photoshop. All calculations were performed in Microsoft Excel and the results are reported as the mean ± standard deviation (error bars). p-values were determined by two-tailed, unpaired t-tests performed in Excel or Graphpad Prism 6.

## RNA-seq analysis

Total RNA was isolated from pancreata of 4- to 5-month-old *Ptf1a* cKO mice (*Ptf1a$^{CreERT/lox}$*) and their corollary controls (*Ptf1a$^{CreERT/+}$*), 2 weeks after TM treatment (3 days, 0.25 mg/g/day), using the guanidine thiocyanate protocol previously described with minor modifications (*MacDonald et al., 1987*). Individual RNA-Seq libraries were prepared from 5 μg of pancreatic RNA from three control and three *Ptf1a* cKO mice with the Illumina True-seq protocol by the UT Southwestern Genomic Core. The sequence data sets from an Illumina HISEQ2500 contained 50-nucleotide uniquely aligned single-end reads of 25.1, 26.4, and 25.8 million for the control samples and 29.7, 24.7, and 26.7 million for the *Ptf1a* cKO RNA samples (Tophat2) (*Kim et al., 2013*). Genes with differential expression were derived using edgeR (*Robinson et al., 2010*), with the default trimmed mean of M-values (TMM) trim settings of 30% for $M_g$ and 5% for $A_g$ and an FDR cut-off of <0.05. The volcano plot of differentially expressed genes was generated using R (http://www.r-project.org/) with the log2 fold change (FC) plotted against the FDR (−log10) (*Supplementary files 1, 2*).

Gene signatures of RAS dependency (*Singh et al., 2009*; *Loboda et al., 2010*), classical and exocrine-like PDAC (*Collisson et al., 2011*), were mapped to orthologous mouse genes via HomoloGene ID. The RAS dependency signature combines gene lists from two separate studies (*Singh et al., 2009*; *Loboda et al., 2010*), comprising 264 genes with only five in common. For GSEA

(*Subramanian et al., 2005*), we analyzed signature enrichment within the entire *Ptf1a* cKO RNA-seq data set, ordered by log2 FC relative to control, using the GSEA desktop application (http://www. broadinstitute.org/gsea/index.jsp).

To identify regulatory pathways altered upon *Ptf1a* deletion, significantly increased and decreased genes were analyzed by IPA (QIAGEN, Redwood City, CA, www.ingenuity.com) at expression thresholds of 1.5-, 2.0,- and 3.0-fold (*Supplementary files 3–5*). In order to obtain an accurate comparison between enriched and downregulated pathways, we used the Comparison Analysis function from expression data filtered at a gene expression threshold of ±2.0-fold. Heat maps were generated according to the −log p-values given by the IPA software using the comparison analysis function and were constructed in R (http://www.r-project.org/).

### Caerulein treatment

We induced acute pancreatitis by i.p. injection of caerulein (Bachem, Torrance, CA), 0.1 μg/g, eight times daily over two consecutive days, as previously (*Jensen et al., 2005*; *Keefe et al., 2012*). Negative controls were injected with saline vehicle alone. Pancreata from all caerulein- or saline-treated mice were harvested 1 week following the final injection and processed as described above.

### Kaplan–Meier analysis

KPC mice (of the genotype *Pdx1-Cre; Kras^{G12D}; p53^{lox/+}*) and KPC mice with *Ptf1a* heterozygosity (*Pdx1-Cre; Kras^{G12D}; p53^{lox/+}; Ptf1a^{Δ/+}*) were aged until they exhibited lethargy or distress as determined by the authors (NMK and LCM) and the in-house veterinary staff, or until the detection of a firm abdominal mass by palpation. The presence of PDAC was confirmed by histological analysis in consultation with a surgical pathologist (MB). At sacrifice, all mice were thoroughly inspected for liver metastases. Survival analysis was performed in GraphPad Prism (Version 6) and p-values were calculated using a Log-rank test.

## Acknowledgements

We are grateful to Matt Firpo, Gabrielle Kardon, and Kirk Thomas for helpful comments on the manuscript, and to Zev Kronenberg and Peter Hendrickson for assistance with bioinformatic analyses. This work was supported by NIH grants P01-DK42502 to CVEW, R01-DK061220 to RJM, and R21-CA179453 to LCM. NMK was a trainee of the University of Utah Developmental Biology Training Grant, T32-HD007491, and is currently supported by NIH fellowship F30-CA192819.

## Additional information

### Funding

| Funder | Grant reference | Author |
| --- | --- | --- |
| National Cancer Institute (NCI) | R21-CA179453 | Nathan M Krah, L Charles Murtaugh |
| National Institute of Diabetes and Digestive and Kidney Diseases (NIDDK) | R01-DK061220 | Raymond J MacDonald |
| National Institute of Diabetes and Digestive and Kidney Diseases (NIDDK) | P01-DK42502 | Christopher VE Wright |
| Eunice Kennedy Shriver National Institute of Child Health and Human Development | T32-HD007491 | Nathan M Krah |
| National Cancer Institute (NCI) | F30-CA192819 | Nathan M Krah |

The funders had no role in study design, data collection and interpretation, or the decision to submit the work for publication.

## Author contributions
NMK, J-PDLO, GHS, Conception and design, Acquisition of data, Analysis and interpretation of data, Drafting or revising the article; CQH, GMC, Acquisition of data, Analysis and interpretation of data, Drafting or revising the article; SGW, FCP, Drafting or revising the article, Contributed unpublished essential data or reagents; MPB, Analysis and interpretation of data, Drafting or revising the article; CVEW, Conception and design, Drafting or revising the article, Contributed unpublished essential data or reagents; RJMD, Conception and design, Acquisition of data, Analysis and interpretation of data, Drafting or revising the article, Contributed unpublished essential data or reagents; LCM, Conception and design, Analysis and interpretation of data, Drafting or revising the article

## Ethics
Animal experimentation: This study was performed according to institutional and National Institutes of Health guidelines for animal research (Guide for the Care and Use of Laboratory Animals), and followed protocols approved by the Institutional Animal Care and Use Committees of the University of Utah (protocol #13-09009), University of Texas Southwestern Medical Center (protocol #2013-0008) and Vanderbilt University (protocol #M10/106).

## Additional files

### Supplementary files
• Supplementary file 1. R markdown for RNA-seq analysis. HTML annotation of R software package analysis performed to generate the data and analyses presented in *Figure 5*.

• Supplementary file 2. Files used for RNA-seq analysis. Excel spreadsheet containing (as tabs) files used in R analysis of RNA-seq data, including differentially expressed genes (false discovery rate < 0.05), gene signatures of RAS dependency, classical pancreatic ductal adenocarcinoma (PDAC) and exocrine-like PDAC, and 'palette' file used for color-coding volcano plot.

• Supplementary file 3. Ingenuity Pathway Analysis (IPA) analysis output for genes changed >1.5-fold. Excel spreadsheet output from IPA (www.ingenuity.com), indicating predicted up- and down-regulated pathways and regulators from *Ptf1a* conditional knock-out (cKO) RNA-seq data, based on a differential expression threshold of 1.5-fold.

• Supplementary file 4. IPA analysis output for genes changed >twofold. Excel spreadsheet output from IPA (www.ingenuity.com), indicating predicted up- and down-regulated pathways and regulators from *Ptf1a* cKO RNA-seq data, based on a differential expression threshold of twofold.

• Supplementary file 5. IPA analysis output for genes changed >threefold. Excel spreadsheet output from IPA (www.ingenuity.com), indicating predicted up- and down-regulated pathways and regulators from *Ptf1a* cKO RNA-seq data, based on a differential expression threshold of threefold.

### Major dataset
The following dataset was generated:

| Author(s) | Year | Dataset title | Dataset ID and/or URL | Database, license, and accessibility information |
| --- | --- | --- | --- | --- |
| Hoang Chinh, Galvin H Swift, Ana Azevedo-Pouly, Raymond J MacDonald | 2015 | Effects on the transcriptome of adult mouse pancreas (principally acinar cells) by the inactivation of the Ptf1a gene in vivo | http://www.ncbi.nlm.nih.gov/geo/query/acc.cgi?acc=GSE70542 | Publicly available at NCBI Gene Expression Omnibus (Accession No: GSE70542). |

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
