## [Decision Letter]

Thank you for sending your work entitled “The acinar differentiation determinant PTF1A inhibits initiation of pancreatic ductal adenocarcinoma” for consideration at *eLife*. Your article has been favorably evaluated by Charles Sawyers (Senior editor), Michael Green (Reviewing Editor), and three expert reviewers that included Narendra Wajapeyee and Marina Pasca di Magliano.

The Reviewing editor and the reviewers discussed their comments before we reached this decision, and the Reviewing editor has assembled the following comments to help you prepare a revised submission.

The manuscript entitled “The acinar differentiation determinant PTF1A inhibits initiation of pancreatic ductal adenocarcinoma” investigates the role of the transcription factor PTF1A in the adult pancreas, including acinar cell maintenance, tissue repair following the induction of pancreatitis and carcinogenesis. The authors find that PTF1A is required to maintain acinar differentiation in adult mice. Intriguingly, loss of *Ptf1a* facilitates the onset of acinar-ductal metaplasia (ADM), and the onset of KRAS-driven pancreatic carcinogenesis. These findings fit into a nascent paradigm that indicates dedifferentiation of acinar cells as a limiting step for pancreatic carcinogenesis. These findings have relevance for our basic understanding of pancreatic acinar biology and carcinogenesis, and have implications for prevention strategies for pancreatic cancer in high-risk populations. The study is carefully designed and executed, and clearly written. It is novel and addresses important open questions, thus should be of interest to a broad readership.

1) The authors use a *Rosa26*-EYFP allele to follow Cre-mediated recombination. This is powerful strategy, that allows detection and sorting of cells that have undergone Cre recombination. However, it is a common finding that recombination at the *Rosa26* locus does not always correspond to recombination at other loci such as, in this case *Ptf1a*. The authors note that recombination in 15-20% of acinar cells is not sufficient to induce ADM, but recombination in a higher percentage of cells does indeed induce ADM, and interpret these findings as a “field” effect, or, in other words, a non-cell autonomous effect of *Ptf1* loss. This is a valid interpretation, however alternative explanations should be tested:

A) It would be important to determine how many cells have effectively lost *Ptf1a* expression both in the low-dose and high-dose tamoxifen experiments, independently from EYFP expression. This could be achieved by co-staining for *Ptf1a*, acinar differentiation markers such as CPA1 and EYFP. This experiment would determine whether retained *Ptf1a* expression prevents ADM in the low-dose experiment.

B) Alternatively, or in addition, the authors could sort the EYFP+ cells and determine the recombination status of the *Ptf1a* locus at the DNA level by PCR.

2) High level-tamoxifen induces ADM while low-level tamoxifen does not. One factor to be excluded is whether high-dose tamoxifen causes transient inflammation that in turn results into ADM. An inflammatory process would similarly explain the field-effect observed. Thus, harvesting mice immediately following tamoxifen administration, and/or treating mice with anti-inflammatory drugs during the tamoxifen treatment, could address this question.

3) 3D acinar cultures could be used to determine the propensity of acinar clusters to undergo ADM in presence or absence of *Ptf1a*. In addition, acinar cultures could be used to determine whether the gene expression changes observed in *Ptf1a* knock-out mice are mainly a reflection of the increased number of lesions, or they are a direct effect of *Ptf1a* inactivation.

4) For the caerulein experiments, the authors should measure the levels of MAPK signaling activation (p-ERK staining) in the ductal lesions. Oncogenic KRAS, and MAPK signaling, have been shown to promote and be essential for de-differentiation of acinar cells. The epistatic relationship between *Ptf1a* and MAPK signaling, if any exists, would be interesting, and clinically relevant given the availability of MEK inhibitors.

5) Figure 3—figure supplement 1: is the stromal response less than control mice in areas with lesions? Or more appropriately, is the stromal response greater in areas of the knockouts without lesions? Or would the authors conclude that the stromal reaction is a response to lesion formation rather than *Ptf1a* deletion?

6) Figure 4. Is the phenotype in the degenerating pancreas of the knockout showing a full blown pancreatitis phenotype (i.e. is it inflamed?)

7) Figure 4. Are the CK19 positive cells YFP positive (distinguish between transdifferentiation and proliferative response), as they are in the cerulein experiments? Is there an enhancement in proliferation or death that accompanies the phenotype?

8) Figure 6: Though other groups have reported that all Alcian Blue positive lesions are a way to distinguish PanIN from metaplasia, this is questionable. To call them PanIN-like is a compromise, but it would be equally legitimate to call Alcian blue-positive PanINs “stable metaplasia-like” lesions (since other models with metaplasia and no PanINs, like the TGFα overexpressers stain positive with Alcian Blue). Do the lesions in Figure 6 stain for Claudin-18 or Muc5ac, more typical PanIN markers? Would a pathologist classify these lesions as metaplasia or PanIN?

---

## [Author Response]

*1) The authors use a* Rosa26*-EYFP allele to follow Cre-mediated recombination. This is powerful strategy, that allows detection and sorting of cells that have undergone Cre recombination. However, it is a common finding that recombination at the* Rosa26 *locus does not always correspond to recombination at other loci such as, in this case* Ptf1a*. The authors note that recombination in 15-20% of acinar cells is not sufficient to induce ADM, but recombination in a higher percentage of cells does indeed induce ADM, and interpret these findings as a “field” effect, or, in other words, a non-cell autonomous effect of* Ptf1 *loss. This is a valid interpretation, however alternative explanations should be tested*:

*A) It would be important to determine how many cells have effectively lost* Ptf1a *expression both in the low-dose and high-dose tamoxifen experiments, independently from EYFP expression. This could be achieved by co-staining for* Ptf1a*, acinar differentiation markers such as CPA1 and EYFP. This experiment would determine whether retained* Ptf1a *expression prevents ADM in the low-dose experiment*.

*B) Alternatively, or in addition, the authors could sort the EYFP+ cells and determine the recombination status of the* Ptf1a *locus at the DNA level by PCR*.

We thank the reviewers for this important comment, which addresses a clear limitation in our conclusions. We have pursued strategy (a), performing quantitative immunofluorescence for *Ptf1a*, the Cre reporter EYFP, and DAPI (Figure 2—figure supplement 2). We used *Ptf1a* cKO + *Kras*^*G12D*^ mice (we do not have adequate numbers of cKO alone mice available, due to our current breeding strategy), treated with no TM, low dose TM or high dose TM, and harvested three days after the final TM treatment. As expected, we found that low-dose TM reduced the fraction of *Ptf1a+* cells per field from ∼80% (in TM -untreated mice) to ∼60%, concomitant with induction of EYFP in ∼20% of cells per field, while high-dose TM reduced *Ptf11a+* cells to ∼15%, with activation of EYFP in ∼50% of all cells. Importantly, in both low- and high-dose TM conditions, the majority of EYFP+ cells had lost *Ptf1a* protein expression (∼75% with low-dose, ∼90% with high-dose). Therefore, EYFP induction does represent a reasonable surrogate for *Ptf1a* deletion (if anything, it may underestimate the extent of *Ptf1a* deletion), and the lack of ADM within EYFP+ cells following low-dose TM is very unlikely to represent a non-deleted *Ptf1a* allele in EYFP+ cells. Instead, we infer that ADM requires a non-cell- autonomous effect produced by widespread *Ptf1a* deletion, consistent with some sort of “field” effect.

*2) High level-tamoxifen induces ADM while low-level tamoxifen does not. One factor to be excluded is whether high-dose tamoxifen causes transient inflammation that in turn results into ADM. An inflammatory process would similarly explain the field-effect observed. Thus, harvesting mice immediately following tamoxifen administration, and/or treating mice with anti-inflammatory drugs during the tamoxifen treatment, could address this question*.

Addressing this important question provided an essential control experiment for our study (Figure 4—figure supplement 3)*.* We divided CD1 wild-type (WT) mice into 4 cohorts and administered the following treatments: no treatment (negative control), high-dose corn oil (vehicle) only, high dose TM dissolved in corn oil (as per our previous experiments), or caerulein administration (6x1 hr i.p. injections at 0.1 μg/g body weight) to induce mild acute pancreatitis as a positive control. Pancreata from all mice were harvested 24 hours post-treatment, and analyzed by histology and immunostaining. We found that pancreata of mice receiving either high-dose corn oil or high-dose TM were indistinguishable from untreated controls, in terms of overall histology as well as frequency of infiltrating CD45+ leukocytes. As expected, caerulein-treated mice exhibited edema and leukocyte infiltration, indicating ongoing inflammation. We therefore conclude that high dose TM treatment alone does not induce pancreatic inflammation, suggesting that the ADM observed upon widespread *Ptf1a* deletion is unlikely to be caused by tamoxifen delivery itself. We have not undertaken anti-inflammatory drug treatment experiments, however, as the range of potential outcomes is wide and we hope to pursue this line of investigation in an independent study.

*3) 3D acinar cultures could be used to determine the propensity of acinar clusters to undergo ADM in presence or absence of* Ptf1a*. In addition, acinar cultures could be used to determine whether the gene expression changes observed in* Ptf1a *knock-out mice are mainly a reflection of the increased number of lesions, or they are a direct effect of* Ptf1a *inactivation*.

We thank the reviewers for this suggestion and we have now performed experiments to address this comment. We prepared acinar cell cultures from mice of the following genotypes: control, *Kras*^*G12D*^, *Ptf1a* cKO, and *Ptf1a* cKO; *Kras*^*G12D*^. Acinar clusters under the influence of *Kras*^*G12D*^ or EGFR signaling have been previously shown to undergo metaplasia into ductal cysts, and we therefore used *Kras*^*G12D*^ pancreata as a baseline to assess metaplasia. While *Ptf1a* cKO pancreata did not robustly form metaplastic spherules in this assay, in the absence of added TGFα, both *Kras*^*G12D*^ and *Ptf1a* cKO; *Kras*^*G12D*^ acinar clusters formed robust cysts without added growth factors. Importantly, the cysts from *Ptf1a* cKO; *Kras*^*G12D*^ were significantly larger than those derived from *Kras*
^*G12D*^ pancreata (Figure 3—figure supplement 2) . Importantly, these results suggest that loss of *Ptf1a* has a direct effect on the ability of Kras^G12D^ to transform acinar cells, independent of inflammation (Figure 6) and other non-cell autonomous pathways (Figure 5). Nice!

*4) For the caerulein experiments, the authors should measure the levels of MAPK signaling activation (p-ERK staining) in the ductal lesions. Oncogenic KRAS, and MAPK signaling, have been shown to promote and be essential for de-differentiation of acinar cells. The epistatic relationship between* Ptf1a *and MAPK signaling, if any exists, would be interesting, and clinically relevant given the availability of MEK inhibitors*.

We thank the reviewer for this request, which has produced an important insight, per their suggestion, into the epistasis of *Ptf1a* and MAPK (Figure 6—figure supplement 1, C-F). Based on phospho-ERK immunostaining, we observe MAPK activation specifically in the ductal lesions of caerulein-treated *Ptf1a* cKO mice, but not in the normal acini of saline-treated cKO mice (nor in control mice, regardless of treatment). This result is consistent with the pancreatitis-induced lesions of *Ptf1a* cKO pancreata being generated via amplification of endogenous RAS signaling; genetically, this implies that *Ptf1a* acts as a negative upstream regulator to MAPK, via biochemical mechanisms to be determined.

*5)*
Figure 3—figure supplement 1*: is the stromal response less than control mice in areas with lesions? Or more appropriately, is the stromal response greater in areas of the knockouts without lesions? Or would the authors conclude that the stromal reaction is a response to lesion formation rather than* Ptf1a *deletion*?

Our SMA staining data suggest that the stromal reaction observed after only two weeks of simultaneous *Ptf1a* deletion and *KRAS*^*G12D*^ activation is a response to lesion formation, rather than a reaction to *Ptf1a* deletion itself. We also observe local SMA staining around lesions in mice with oncogenic *KRAS*^*G12D*^ alone, suggesting that this is not a specific reaction to *Ptf1a* loss, per se, but to PanIN formation. Nonetheless, our gene expression profiling provides evidence that stellate cell signaling is upregulated in response to *Ptf1a* deletion (Figure 5), suggesting that *Ptf1a* deletion results in a priming signal to stellate cells that enhances their activation when other insults, such as *KRAS*^*G12D*^ activation, drive acinar cell transformation. We now make this point within the text (subsection “Loss of *Ptf1a* expression is a rate-limiting step for PanIN initiation”).

*6)*
Figure 4*. Is the phenotype in the degenerating pancreas of the knockout showing a full blown pancreatitis phenotype (i.e. is it inflamed?)*

This is an important concern, given that gene expression profiling suggests upregulation of fibro-inflammatory pathways upon *Ptf1a* deletion (Figure 5). To examine this directly, we stained pancreata of control and *Ptf1a* cKO mice, 2 weeks after high-dose TM, for EYFP, CK19, and the leukocyte marker CD45, as well as pancreata of caerulein-treated mice as a positive control (Figure 4—figure supplement 2). While CD45 staining is robust in mice treated with caerulein, we observed minimal CD45 staining in *Ptf1a* cKO pancreata. Importantly, CD45+ cells within *Ptf1a* cKO pancreata localized specifically to areas of ongoing ADM, indicated by the presence of EYFP+/CK19+ cells. These data are consistent with previous studies in which ADM is accompanied by and, in some cases, driven by interactions with inflammatory cells such as macrophages. As noted above, we plan to address this question in a future study. However, we conclude that the loss of *Ptf1a* does not confer a full-blown pancreatitis phenotype, but instead creates the potential for locally-enhanced inflammation (which is dramatically realized upon caerulein treatment).

*7)*
Figure 4*. Are the CK19 positive cells YFP positive (distinguish between transdifferentiation and proliferative response), as they are in the cerulein experiments? Is there an enhancement in proliferation or death that accompanies the phenotype*?

We find that aberrant CK19+ structures do arise from EYFP+ acinar cells, although they are relatively rare. We have addressed the issues raised in this valuable comment by analyzing control and *Ptf1a* cKO pancreata for transdifferentiation, cell death and proliferation (Figure 4—figure supplement 2). Altogether, our results suggest that widespread *Ptf1a* deletion is accompanied by both transdifferentiation and proliferation. Interestingly, as noted above, we find that many transdifferentiated (CK19+) acinar cells are in close proximity to CD45+ leukocytes, reinforcing previous studies indicating interdependence of metaplasia and inflammation.

*8)*
Figure 6*: Though other groups have reported that all Alcian Blue positive lesions are a way to distinguish PanIN from metaplasia, this is questionable. To call them PanIN-like is a compromise, but it would be equally legitimate to call Alcian blue-positive PanINs “stable metaplasia-like” lesions (since other models with metaplasia and no PanINs, like the TGFα overexpressers stain positive with Alcian Blue). Do the lesions in*
Figure 6
*stain for Claudin-18 or Muc5ac, more typical PanIN markers? Would a pathologist classify these lesions as metaplasia or PanIN*?

This is an important question and we have discussed it extensively within our group. While lesions in the *Ptf1a* cKO + caerulein mouse are almost ubiquitously Alcian blue reactive, very few stain weakly with Claudin18 and most do not stain at all. There is a subset of lesions that are weakly Claudin-18 positive, but this staining is not as robust as PanINs formed via *Kras*^*G12D*^ activation stained in parallel (Figure 6—figure supplement 1). A similar trend is noted for MUC5ac. Our pathologist co-author, Dr. Bronner, indicates that these lesions cannot be classified as PanINs based on histology, but are rather metaplastic lesions with varying degrees of dysplasia. We therefore refer to these structures as “mucinous metaplastic structures” in the text, rather than “PanIN-like” (please see the subsection “Caerulein-induced pancreatitis is sufficient to reprogram *Ptf1a*-deficient acinar cells”).